# Elaboration of Nanostructured Levan-Based Colloid System as a Biological Alternative with Antimicrobial Activity for Applications in the Management of Pathogenic Microorganisms

**DOI:** 10.3390/nano13222969

**Published:** 2023-11-17

**Authors:** Vitalijs Radenkovs, Anda Valdovska, Daiga Galina, Stefan Cairns, Dmitrijs Jakovlevs, Sergejs Gaidukovs, Ingmars Cinkmanis, Karina Juhnevica-Radenkova

**Affiliations:** 1Processing and Biochemistry Department, Institute of Horticulture, LV-3701 Dobele, Latvia; karina.juhnevica-radenkova@llu.lv; 2Research Laboratory of Biotechnology, Latvia University of Life Sciences and Technologies, LV-3004 Jelgava, Latvia; anda.valdovska@lbtu.lv (A.V.); daigag@llu.lv (D.G.); dmitrijs.jakovlevs@gmail.com (D.J.); 3Faculty of Veterinary Medicine, Latvia University of Life Sciences and Technologies, LV-3004 Jelgava, Latvia; 4Malvern Panalytical Ltd., Worcestershire, Malvern WR14 1XZ, UK; 5Institute of Polymer Materials, Faculty of Materials Science and Applied Chemistry, Riga Technical University, LV-1048 Riga, Latvia; sergejs.gaidukovs@rtu.lv; 6Faculty of Agriculture and Food Technology, Latvia University of Life Sciences and Technologies, LV-3004 Jelgava, Latvia; ingmarsc@lbtu.lv

**Keywords:** biopolymers, exopolysaccharides, healing, levan, multi-drug resistance, nanoparticles, wounds

## Abstract

Considering the documented health benefits of bacterial exopolysaccharides (EPSs), specifically of bacterial levan (BL), including its intrinsic antimicrobial activity against certain pathogenic species, the current study concentrated on the development of active pharmaceutical ingredients (APIs) in the form of colloid systems (CoSs) containing silver nanoparticles (AgNPs) employing in-house biosynthesized BL as a reducing and capping agent. The established protocol of fermentation conditions implicating two species of lactic acid bacteria (LAB), i.e., *Streptococcus salivarius* K12 and *Leuconostoc mesenteroides* DSM 20343, ensured a yield of up to 25.7 and 13.7 g L^−1^ of BL within 72 h, respectively. An analytical approach accomplished by Fourier-transform infrared (FT-IR) spectroscopy allowed for the verification of structural features attributed to biosynthesized BL. Furthermore, scanning electron microscopy (SEM) revealed the crystalline morphology of biosynthesized BL with a smooth and glossy surface and highly porous structure. Molecular weight (M_w_) estimated by multi-detector size-exclusion chromatography (SEC) indicated that BL biosynthesized using *S. salivarius* K12 has an impressively high M_w_, corresponding to 15.435 × 10^4^ kilodaltons (kDa). In turn, BL isolated from *L. mesenteroides* DSM 20343 was found to have an M_w_ of only 26.6 kDa. Polydispersity index estimation (PD = M_w_/M_n_) of produced BL displayed a monodispersed molecule isolated from *S*. *salivarius* K12, corresponding to 1.08, while this was 2.17 for *L*. *mesenteroides* DSM 20343 isolate. The presence of fructose as the main backbone and, to a lesser extent, glucose and galactose as side chain molecules in EPS hydrolysates was supported by HPLC-RID detection. In producing CoS-BL@AgNPs within green biosynthesis, the presence of nanostructured objects with a size distribution from 12.67 ± 5.56 nm to 46.97 ± 20.23 was confirmed by SEM and energy-dispersive X-ray spectroscopy (EDX). The prominent inhibitory potency of elaborated CoS-BL@AgNPs against both reference test cultures, i.e., *Pseudomonas aeruginosa*, *Escherichia coli*, *Enterobacter aerogenes*, and *Staphylococcus aureus* and those of clinical origin with multi-drug resistance (MDR), was confirmed by disc and well diffusion tests and supported by the values of the minimum inhibitory and bactericidal concentrations. CoS-BL@AgNPs can be treated as APIs suitable for designing new antimicrobial agents and modifying therapies in controlling MDR pathogens.

## 1. Introduction

Chronic wounds are an immense burden for the individual patient and society [1]. The impact of wounds is referred to as “the silent epidemics” as it is felt on many levels, starting from lost work days to permanent disability (amputation) to death of the patient, and the economic implications of chronic wound care are alarming [2,3]. Among global threats in this century, the relevance of controlling and suppressing the spread of microbial infections has become the number one challenge. Overuse of antibiotics, which is a primary driver leading to the development of antimicrobial resistance (AMR) [4], along with governmental intentions outlined in the new European Commission (EC)’s Regulation No. 2019/6 [5], make investigators from around the world seek alternative strategies to reduce this need both in human and veterinary medicine [6]. Despite the success of the last decades in the treatment of hard-to-heal wounds (H–HW) both in human [7] and veterinary patients [8], there is a need for designing new antimicrobial agents and modifying treatments feasible in combating multi-drug-resistant (MDR) pathogens [9,10]. In non-healing wounds, wound chronicity is associated with the ability of bacteria to form a biofilm composed of bacterial aggregates by producing a polymer-based matrix of a polysaccharide/protein nature [11]. The film-forming capabilities of bacteria make them more resistant to topical antibiotics and biocides within therapeutic maneuvers [12]. Repeated attacks on a regular basis after 48–96 h of biofilm formation by applying disruptive strategies such as debridement force the biofilm to reattach and reform, making it sensitive to antibiotics and host defenses [13]. The advancement in molecular biology, genetics, and diagnostics allows for the routine performance of robust identification and differentiation of pathogenic bacteria and infectious diseases, giving a chance to appoint effective antibiotic therapy without running the risk of biofilm formation [14,15]. However, the development of MDR pathogens and chronic inflammation due to the lack of equipment available in its place, along with health issues attached to long-distance travel, lead to delayed wound healing, particularly in canine and equine limbs. The transition of a wound from an acute into a chronic form is accompanied by a high level of inflammatory cytokines, pronounced protease activity, increased levels of reactive oxygen species, degraded and nonfunctional extracellular matrix (ECM) reorganization, and aged cells with low mitogenic activity [16]. In wound management, the presence of these factors, along with drug implications such as corticosteroids, chemotherapeutics, immunosuppressive agents, and nonsteroidal anti-inflammatory drugs (NSAIDs), make the wound healing process long-lasting [17,18]. Meanwhile, the biofilm formation that was identified in ≤60% of chronic wound cases of cattle and horses, mainly due to multiple bacterial contributions, makes antibiotic therapy less effective [19,20].

To date, efficient biological solutions for treating H–HW and ensuring a prolonged antibacterial activity are still limited. Although, such solutions can become quite realistic if they involve microbial exopolysaccharides (EPSs). EPSs are metabolites of polysaccharides produced and secreted by a variety of microorganisms, including lactic acid bacteria (LAB) [21]. Findings in the area of biotechnology driven in recent years have the prospect of conceiving plenty of EPSs alone or in combination with APIs or antimicrobial agents, benefitting multiple functionalities and a range of applicability [17,18], and this field has attracted steadily growing interest among researchers and inventors worldwide [22]. The interest in EPSs, to a greater extent, is conditioned by their amphiphilic nature and ability to interact with bacterial membranes, leading to the alteration of glycoproteins and lipids on the cytomembrane of microorganisms, thus affecting ion exchange and reducing ATF synthesis. This statement can be reinforced by an observation made by Goy et al. [23] for chitosan and, recently, by Aullybux et al. [24] for EPSs isolated from marine bacteria *Mauritius seawater*. Antimicrobial and antibiofilm efficiency of bacterial cellulose has been constantly documented [25,26]. However, of the many existing EPSs, bacterial levan (BL) is another EPS representative that demonstrates unique intrinsic antimicrobial activity against pathogenic microorganisms, as reported by Zainulabdeen et al. [27] and Hamada et al. [28]. However, Gökmen et al. [29] emphasized the opacity of BL antimicrobial activity against certain bacterial species, such as *Salmonella typhimurium*, assumed to be the cause of insufficient purity of BL and the presence of other less active metabolites extracted along with BL. Regardless, the authors proposed using BL as an adjuvant to other active molecules to promote antimicrobial activity. BL is a natural β-D-fructofuranose polymer cross-linked by β-(2→6) glycosidic linkages at the C-1 position of the fructose ring [30,31]. BL is a water-soluble, strongly adhesive, and film-forming homopolysaccharide, and it differs from other EPSs in its unique properties, such as its low viscosity, high oil solubility, compatibility with salts and surfactants, thermo-, acid, and alkaline resistance, and high water retention capacity [31]. To achieve augmentation benefits in combating MDR and persistent pathogens along with the wound repair process, González-Garcinuño et al. [32] proposed using BL in preparing nanostructured BL@AgNP hydrogel with remarkable inhibitory activity against *Escherichia coli* and *Bacillus subtilis*. Using BL in the green biosynthesis of Ag nano-objects as a reducing agent ensured the formation of BL@AgNPs with particles of about 30 nm. Although Khan et al. [33] elaborated upon functionalized gold (Au) and AgNPs with biomedically active functional groups inherited from *Clerodendrum inerme*, ensuring superior antibacterial and antimycotic activities against seven species of pathogenic microorganisms along with biofilm inhibition capability, it is worth noting, though, that the use of crude extracts in metal NP biosynthesis could lead to the appearance of undesirable complexes in the final products resulting in limiting the range of AgNP applicability [8]. Therefore, more attention has recently been paid to synthesizing nano-scale objects using individual components. In this way, encouraging results have been obtained by Ahmed [34], demonstrating the potential utilization of BL as a sole reducing agent in the biosynthesis of AgNPs with remarkable catalytic activity in the reduction of 4-nitrophenol to methylene blue. Choi et al. [35] revealed an injectable, non-cytotoxic, and cell-proliferative BL-based hydrogel as a dermal filler for soft tissue. Observations made by the researchers provide a platform to combating *E. faecium*, *S. aureus*, *K. pneumoniae*, *A. baumannii*, *P. aeruginosa*, and *Enterobacter* spp. (ESKAPE) in the era of AMR [36,37,38].

Considering the health-promoting evidence about BL and already established protocol for BL production via a fermentation process utilizing LAB, i.e., *S. salivarius* K12 and *L. mesenteroides* DSM 20343 cultures developed by a group from LatHort, the current research will address the chromatographic characterization of obtained BL and elucidation of its potential contribution to the formation of AgNPs during green biosynthesis. As a complement, the antimicrobial activity of prepared CoS-BL@AgNPs will be probed against certain pathogenic bacteria with multi-drug resistance.

## 2. Materials and Methods

### 2.1. Chemicals, Reagents, Standards

A commercial standard of dextran (DEX) derived from *L. mesenteroides* with an average molecular weight, M_w_, of 1500–2800 kilodaltons (kDa), sodium hydroxide, citric acid, and absolute ethanol (purity ≥ 99.5%) were purchased from Merck KGaA (Darmstadt, Germany). An acetonitrile (MeCN) and methanol (MeOH) gradient grade for liquid chromatography (HPLC) was purchased from the same producer. Standards of mono- and disaccharides, dipotassium hydrogen phosphate, manganese sulfate, magnesium sulfate, and Tween^®^ 80 were purchased from Sigma-Aldrich (Steinheim, Germany). Sucrose and sodium acetate were obtained from MilliporeSigma (Rockville, MD, USA). The diammonium citrate used was from Scharlab S.L., (Barcelona, Spain). Dimethyl sulfoxide (DMSO) of a liquid chromatography–mass spectrometry (LC–MS) grade (purity ≥ 99.7%) was obtained from Thermo Fisher Scientific (Rockford, IL, USA). Silver nitrate (AgNO_3_) (purity ≥ 99.9%) and ammonium hydroxide solution (25% *v*/*v*) were obtained from Chempur (Piekary Śląskie, Silesia, Poland). *Streptococcus salivarius* K12 test culture was isolated from the oral cavity and successfully identified using standard morphological and biochemical analysis and PCR technique, while *L. mesenteroides* DSM 20343 was obtained from the collection of the UK Health Security Agency (UKHSA) upon request. Mueller–Hinton broth (MHB) was purchased from Oxoid, Inc., Thermo Fisher Scientific (Hampshire, UK). De Man, Rogosa, and Sharpe (MRS) agar supplemented with Tween^®^ 80 and Mueller–Hinton Agar (MHA) II were acquired from Biolife Italiana S.r.l. (Milan, Italy). Bacteriological peptone (pH 6.5–7.5) and yeast extract (pH 6.8–7.2) were purchased from Laboratorios Conda S.A (Madrid, Spain) and beef extract (pH 7.0 ± 0.5) was obtained from Liofilchem^®^ S.r.l (Roseto, Italy). The deionized water (DeW) was obtained from the reverse osmosis of the “PureLab Flex Elga” purification system (Veolia Water Technologies, Paris, France).

### 2.2. Biosynthesis of Bacterial Levan In Vitro Using Streptococcus salivarius K12 and Leuconostoc mesenteroides DSM 20343

Pure LAB test cultures were cultivated aerobically at 37 ± 1 °C for 48 h on non-selective MRS agar containing the following (in g L^−1^): bacteriological peptone, 20.0; beef extract, 10.0; yeast extract, 5.0; D-glucose, 20.0; K_2_HPO_4_, 2.0; C_2_H_3_NaO_2_, 5.0; C_6_H_14_N_2_O_7_, 2.0; MgSO_4_, 0.2; MnSO_4_ 0.05; and C_64_H_124_O_26_, 1.0, pH adjusted to 6.9, when appropriate media was solidified with 15.0 g L^−1^ agar. Production of BL was implemented following the protocol (ME.PL.011.2021) developed by a group at LatHort. Briefly, 27.6 g L^−1^ of modified MRS broth (Table 1) was suspended in 500 mL of deionized water (DW) in 2 L autoclavable reagent bottles with a screw cap (VWR^™^ International, GmbH, Darmstadt, Germany) and the bottles’ content was subjected to autoclaving using a “Raypa, AES 110” (Barcelona, Spain) digital autoclave with counter-pressure of 2.0 Pa for 15 min at 121 ± 1 °C. After the prepared broth had reached ambient temperature, a 5 mL suspension containing the reactivated microorganism culture with a final turbidity of 3.0–4.0 McFarland units (10^8^–10^9^ CFU mL^−1^) was introduced to the broth. In parallel, inoculation (using MRS agar) with the subsequent plate counting technique was performed to confirm the concentration of the pure culture in the suspension inoculated to the broth for the biosynthesis of BL. The fermentation lasted 72 h at 37 ± 1 °C and 60 rpm. The reaction was terminated by subjecting the obtained fraction to thermal processing for 10 min at 99 ± 1 °C.

To separate the biomass and eliminate bacterial remains, the obtained fraction rich in BL was centrifuged at 10,280× *g* for 10 min at 0 ± 1 °C, using a laboratory-refrigerated centrifuge, “Hermle Z 36 HK” (Hermle Labortechnik, GmbH, Wehingen, Germany). The obtained top layer was collected in 2 L autoclavable reagent bottles with a screw cap. Afterwards, the chemical precipitation of BL was performed using cooled absolute ethanol 99.5% (EtOH) (BL levan fraction:EtOH ratio 1:3 *v*/*v*) for 48 h at 4 ± 1 °C. The precipitation of BL diluted in DeW was repeated 3 times to achieve the desired purity of produced BL. Afterwards, the recovered sediment fraction was subjected to centrifugation at 10,280× *g* for 10 min at 0 ± 1 °C and freeze-drying using a “HyperCOOL, HC3110” freeze-drying system (Hanil Scientific Inc., Gimpo, Republic of Korea) at −50 ± 1 °C under a vacuum of 0.056–0.070 mBar for 24 h. The obtained dry BL was ground to reach a Ø 0.2 mm particle size using a laboratory ball mill, “Anton Paar BM500” (Anton Paar GmbH, Graz, Austria), at a frequency of 15 Hz for 2 min. A schematic representation of the BL production steps is depicted in Figure 1.

### 2.3. Preparation of Silver Nanoparticles Using Bacterial Levan Produced by Streptococcus salivarius K12 and Leuconostoc mesenteroides DSM 20343

The stock solution with a molarity of 100 mM AgNO_3_ was applied as the source of Ag. For the preparation of AgNPs using synthetic DEX (control), 50.0 and 500.0 mg of commercially available DEX with molecular weights, M_w_, of 1500–2800 kilodaltons (kDa) were placed in 50.0 mL reagent bottles with a screw cap and heated for 5 min at 80 ± 1 °C with continuous agitation at 500 rpm using a magnetic stirrer, “Arex-6 Digital Pro” (Velp^®^ Scientifica, Usmate Velate, Italy). Subsequently, 100.0 µL of 1.0 M NaOH was introduced through the upper opening, and the solution was allowed to mix well. Later, 100.0 µL of 100 mM AgNO_3_ solution was introduced drop-wise, thus providing 1 mM AgNO_3_ in the final solution. In this research, NaOH was employed as an accelerator in the green biosynthesis of BL@AgNPs, as reported by Radenkovs et al. [39]. Since fructose and fructose-type EPS BL are considered to be non-reducing sugar, their application in the green biosynthesis of AgNPs has not been reported so far. However, according to Lobry de Bruyn–Alberda van Ekenstein transformation [40], fructose can be converted into epimers of D-glucose and D-mannose under alkaline conditions within enediol rearrangements and epimerization, and the final products may react with AgNOs as reducing agents. Following the Lobry de Bruyn–Alberda van Ekenstein transformation [40], the first step in glucose conversion is the removal of the proton alpha in ketone under alkaline conditions. During the second step, the resulting enolate is subjected to protonation on oxygen, leading to enediol formation with two hydroxyl (–OH) groups. Further steps involving the enediol rearrangement and deprotonation of a stereocenter carbon atom at two faces lead to the formation of either D-glucose or epimer D-mannose (not shown). However, the final reaction products could be considered a mixture of D-glucose, D-fructose, and D-mannose. In the subsequent reaction, glucose and mannose, as reducing sugars containing a free aldehyde group at the C-1 position, react with Ag^+^, reducing it to Ag^0^. In this process, BL restricts the formation of large, irregular NPs acting as capping/anticoagulation agents similar to those reported for DEX [41].

To elucidate the concentration and the type of BL used as a reducing agent on NP geometry and particle size distribution, the biosynthesis of BL-AgNPs was appraised, varying in the concentration and source of BL (from 20.0 to 500.0 mg). The reaction mixtures were heated for 1 h at 80 ± 1 °C, ensuring constant stirring at 500 rpm. The mixture gradually shifted from light yellow to reddish brown color during thermal exposure over 1 h. The whole reaction was carried out in the dark, covering the vessel with foil to preserve the AgNO_3_ from aggregation. For antimicrobial analysis, part of the obtained BL-AgNP CoS was delivered to the Faculty of Veterinary Medicine, Latvia University of Life Sciences and Technologies. In contrast, the second part was centrifuged at 10,280× *g* for 50 min at 20 ± 1 °C to obtain dry BL@AgNPs for morphological, elemental, and structural analyses by scanning electron microscopy (SEM), energy-dispersive X-ray (EDX) spectroscopy, and Fourier-transform infrared (FT-IR) spectroscopy, respectively. Before freeze-drying, the pellet containing BL@AgNPs was washed 3–4 times with DeW to remove free silver ions. Freeze-drying of BL@AgNPs was conducted at −50 ± 1 °C under a vacuum of 0.056–0.070 mBar for 24 h. Freeze-dried NPs were stored in a cool, dry, and dark place and further used for their characterization. A schematic representation of the CoS-BL@AgNP biosynthesis steps following green chemistry principles is shown in Figure 2.

### 2.4. Preparation of Bacterial Levan for Free and Total Saccharide Analysis

Extraction of free mono- and disaccharides from a dry and ground matrix of BL was carried out by subjecting the sample to gentle heating at a temperature of 60.0 °C for 30 min with further ultrasonic treatment at 50 kHz, an output power of 360 W for 30 min, and a temperature of 25.0 ± 1 °C using an “Ultrasons” ultrasonic bath (J.P. Selecta^®^, Barcelona, Spain). For the extraction of saccharides, the sample (100.0 mg ± 1.0) was mixed with 1.0 mL 50% MeCN (H_2_O:CH_3_CN, *v*/*v*), and the extraction was carried out in 15.0 mL conical plastic tubes (Sarstedt AG & Co. KG, Numbrecht, Germany). After treatment, the resulting mixture was intensively vortexed with a “ZX3” vortex mixer (Velp^®^ Scientifica, Usmate Velate, Italy). For the separation of fractions, the prepared samples were centrifuged at 3169× *g* for 10 min^−1^ and a temperature of 19.0 ± 1 °C using a “Sigma, 2-16KC” centrifuge (Osterode near Harz, Germany). Before HPLC-RID analysis, the collected supernatant was filtered through a 0.45 µm polytetrafluoroethylene (CHROMAFIL^®^ Xtra H-PTFE) hydrophilized membrane filter (Macherey-Nagel GmbH & Co. KG, Düren, Germany).

To specify the composition of BL, the produced EPSs were subjected to mild acidic hydrolysis utilizing hydrochloric (HCl, 0.1, 0.2, 0.5 M), trichloroacetic (TCA, 1.0 M), and oxalic (0.05 M) acids, as proposed by Semjonovs et al. [30]. For this purpose, the sample (100.0 mg ± 1.0) was mixed with 2.0 mL of acid, whose molarities of which are mentioned above, in a 22.0 mL glass Headspace (PerkinElmer, Inc., Waltham, MA, USA) with screw caps and silicone seals and vortex mixing for 1 min. Then, the prepared mixture was subjected to thermal processing in a drying cabinet, “Pol-Eko Aparatura” SP.J. (Wodzisław Śląski, Poland), at a temperature of 90 ± 2 °C either for 3 or 6 h. After hydrolysis, the volume of hydrolysate was brought to 4.0 mL with MeCN, and the pH was adjusted to 6.5–6.8 using 25% NH_4_OH solution if necessary. The final volume was 5.0 mL. Subsequent steps of the sample preparation are identical to those indicated above.

### 2.5. The HPLC-RID Conditions for Carbohydrate Analysis

Quantitative analysis of free and bound mono- and disaccharides in dried BL was performed using a “Waters Alliance” HPLC system (Model No. e2695) coupled to a 2414 RI detector and a 2998 column heater (Waters Corporation, Milford, MA, USA) following the methodology described by Radenkovs et al. [42].

### 2.6. Preparation of Bacterial Levan for Determination of Molecular Weight and Rheological Features

SEC is a solution-based technique in which dissolved macromolecules are separated by size; dissolution studies were carried out to determine with which solvent to undertake the analysis. Sample solutions were prepared using three solvents, i.e., 0.1 M aqueous NaNO_3_ solution, 0.1 M LiBr in dimethyl sulfoxide (DMSO), and 0.1 M LiBr in dimethyl formamide (DMF). The sample solutions were stirred and heated at 80 ± 2 °C overnight for 16 h. It was noted that the samples dissolved more easily in DMSO, as high turbidity in the case of NaNO_3_ and particulates in the case of DMF were observed after mixing solvents with obtained EPSs. The DMSO sample solutions were filtered through a 0.45 µm PTFE membrane before being analyzed using the “OMNISEC” system (Malvern Panalytical Ltd., Malvern, UK).

### 2.7. The Analytical Multi-Detector Gel Permeation Chromatography/Size-Exclusion Chromatography–Liquid Chromatography (GPC/SEC–HPLC) System Conditions for Bacterial Levan Analysis

The analyses of weight average molecular weight (M_w_), number average molecular weight (M_n_), dispersity (Ð), hydrodynamic radius (R_h_), intrinsic viscosity (IV), and recovery of BL were carried out using an “OMNISEC” multi-detection size-exclusion chromatographic (SEC) system (Malvern Panalytical Ltd.). The detectors utilized for these studies were the static light scattering, differential refractive index (RI), and viscometer (Malvern Panalytical Ltd.). Two observation angles, i.e., a right angle (90°) and low angle (7°), were used to acquire the sample chromatograms within the static light-scattering detector. A sample of 100.0 μL was injected onto a styrene-divinyl benzene gel permeation analytical column (GPC) (T6000Ms, 10.0 μm, 300 × 8.0 mm; Malvern Panalytical Ltd.) operating at 50 °C and a flow rate of 0.6 mL min^−1^. Chromatographic separation of analytes was achieved using an isocratic flow rate of a mobile phase composed of DMSO (LiBr 0.1 M). During this work, the autosampler and flow cell temperature were maintained at a constant 25 °C. Data were acquired and processed using “OMNISEC” V11.30+. Detector constants, detector offsets, and band broadening were determined using a pullulan standard (Malvern Panalytical Ltd.). Commercial DEX with a molecular mass ranging from 6 to 2560 kDa (Sigma, St. Louis, MO, USA) was used as the standard.

### 2.8. Scanning Electron Microscope (SEM) and Energy-Dispersive X-ray Spectroscope (EDX)

The morphology of two BLs biosynthesized by *S. salivarius* K12 and *L. mesenteroides* DSM 20343 and engineered into BL@AgNPs in CoSs were observed using a “Mira3” scanning electron microscope (SEM) by “Tescan Orsay Holding”, a.s. (Brno-Kohoutovice, Czech Republic). Freeze-dried BL and BL@AgNPs were mounted in a thin layer onto SEM pin stubs using double-sided adhesive carbon discs, and non-fixed particles were blown by a gentle stream of N_2_. Before SEM observation, BL was coated with a gold–palladium alloy using the sputter coater “Leica EM ACE600” (Leica Microsystems, Wien, Austria). The conditions were adjusted to work under a high vacuum mode utilizing a back-scattered electron (BSE) and secondary electron (SE) detector mixer. The BLs were analyzed by increasing magnification up to 5.0 kx, while BL@AgNPs, by 250.0 kx, for precise dimensional measurement and elemental composition analysis operating at a 15 kV acceleration voltage. Elemental composition analysis of the BL@AgNPs was carried out using an energy-dispersive X-ray spectrometer (EDX) equipped with an “INCA x-act LN2-free Analytical Silicon Drift Detector” with PentaFET^®^ Precision by Oxford Instruments Inc. (Bognor Regis, UK), operating from a 4 to 15 mm working distance for energy spectrum accumulation.

### 2.9. Fourier-Transform Infrared Spectroscopy (FT-IR)

The functional groups of dry BLs biosynthesized by *S. salivarius* K12 and *L. mesenteroides* DSM 20343 and prepared into CoS-BL@AgNPs using BL as a natural reducing and capping agent were analyzed using an “IR-Tracer-100” Fourier-transform infrared spectrophotometer (Shimadzu Corporation, Tokyo, Japan). This technique was reported to be helpful by allowing for the identification of BL due to the peculiar absorption band maxima in the mid-infrared region, which could be distinguished [30]. The 1 mg of dry powder was mixed with 100 mg of potassium bromide (KBr) and pressed to obtain a metal mold. The spectrum was acquired within the infrared region of 400–4000 cm^−1^ with a resolution of 4 cm^−1^. Data were acquired using LabSolutions IR software (Tokyo, Japan), which was also used for instrument control and data processing.

### 2.10. In Vitro Susceptibility Tests

#### 2.10.1. Minimum Inhibitory Concentration (MIC)

The minimum inhibitory concentration (MIC) of CoSs containing either BL@AgNPs or DEX@AgNPs was determined using the microdilution method in 96-well plates and utilizing four reference test cultures, including *Pseudomonas aeruginosa* ATCC101 45, *Staphylococcus aureus* ATCC6538, *Escherichia coli* ATCC25922, and *Enterococcus aerogenes ATCC 13048*, and four bacterial isolates from clinical samples with MDR, including *P. aeruginosa, S. epidermidis*, *E. coli*, and *E. faecium*, following protocol reported by Radenkovs et al. [42]. Pathogenic microorganisms were cultivated on nutrient agar (NA; Oxoid, Inc., Thermo Fisher Scientific, Hampshire, UK) for 20 ± 2 h at 36 ± 1 °C and re-suspended in MHB. The suspension turbidity was adjusted to 0.5 McFarland units (10^8^ colony-forming units (CFU) mL^−1^) by applying a “DEN-1B McFarland Tube Densitometer” (Grant Instruments Ltd., Cambridge, UK). The suspension was subsequently diluted to a concentration of approx. 1.5 × 10^6^ CFU mL^−1^ (bacterial inoculum). Afterwards, a serial two-fold dilution method was used to determine the concentration-dependent antimicrobial potency. For this purpose, the concentration of each CoS in the first well was adjusted to 25% (83.3 µL mL) by mixing 100 µL of either BL@AgNPs or DEX@AgNPs with 100.0 µL of MHB and 100 µL of prepared bacterial inoculum. Afterwards, 100 µL of the first well was diluted with 50 µL of MHB and 50 µL of prepared bacterial inoculum and was introduced to the second well. The negative control consisted of MHB, while the positive control, of MHB and bacterial inoculum with test cultures. To calculate MBC, 1 µL of each positive control aliquot before incubation of the microplate was inoculated into two Mueller–Hinton Agar II plates for the baseline concentration of the tested bacteria. After incubation for 24 h at 36 ± 1 °C, the MIC was defined as the lowest concentration that inhibited bacterial growth based on the absence of visible turbidity [42].

#### 2.10.2. Minimum Bactericidal Concentration (MBC)

The MBC was complementary to the MIC assay to identify the lowest concentration of produced CoS-BL@AgNPs leading to death by bacteria. For this purpose, an aliquot of 1 µL from all the wells showing no visible bacterial growth was subcultured into MHA and incubated for 24 h at 36 ± 1 °C. Subsequently, the CFUs were enumerated, and the MBC was specified as the lowest concentration of BL@AgNPs, leading to 95% bacterial death.

#### 2.10.3. Agar Disc Diffusion Method

The antibacterial activity of elaborated CoSs containing either BL@AgNPs or DEX@AgNPs was probed against selected reference test cultures and clinical isolates with MDR using the Kirby–Bauer disc diffusion test following the protocol provided by Radenkovs et al. [42] with modifications. Briefly, a sterile cotton swab spread the bacteria strains in a concentration of 0.5 McFarland units on the MHA. Then, the sterile paper discs with a diameter of 6 mm were placed on the agar plate and soaked with 50 μL of prepared CoSs composed of either BL@AgNPs or DEX@AgNPs. The plates were incubated for 24 h at 36 ± 1 °C, and the zone of inhibition was observed and measured in mm.

#### 2.10.4. Agar Well Diffusion Method

Due to the negligible diffusivity of certain NPs through the culture media, as reported by Kourmouli et al. [43], under a separate trial, the agar well diffusion method was used to elucidate the antimicrobial potency of the prepared CoS. For this purpose, wells with a diameter of 6 mm were punched in the agar and filled with the same antimicrobials described above. The zone of inhibition was observed and measured in mm after incubation for 24 h at 36 ± 1 °C.

### 2.11. Statistical Analysis

The data acquired are shown as the means ± standard deviations of three replicates (*n* = 3). A *p*-value of ≤0.05 was used to indicate significant differences between mean values established by one-way analysis of variance (ANOVA) and Duncan’s multiple range test conducted using IBM^®^ SPSS^®^ Statistics version 20.0 (SPSS Inc., Chicago, IL, USA).

## 3. Results and Discussion

### 3.1. Characterization of Bacterial Levan Biosynthesized by Streptococcus salivarius K12 and Leuconostoc mesenteroides DSM 20343 Using Fourier-Transform Infrared Spectroscopy

FT-IR analysis was executed to clarify the deposition of functional groups in BL produced by the *S*. *salivarius* K12 and *L*. *mesenteroides* DSM 20343 test cultures. Figure 3 shows the FT-IR spectra of EPSs, while Table 2 summarizes the absorption band assignments. The ring and stretching vibrations were recorded in transmittance mode from the wavenumber of 4.000 to 400 cm^−1^ and compared with the commercially available high-molecular-weight DEX and with spectra of BL available in the literature. Figure 3A represents a broad and strong peak at 3600–3200 cm^−1^, specifically, at 3419 cm^−1^ in *S*. *salivarius* K12 and *L*. *mesenteroides* DSM 20343 attributed to the *ν*O–H stretching vibration due to intermolecular hydrogen bonding [44]. The same stretching vibration of *ν*O–H was observed for fructose (Figure 3B) and supported by Ibrahim et al. [45]. The deposition of *ν*O–H group in DEX was observed at 3404 cm^−1^ and is consistent with data reported by Glišić et al. [46]. The broad asymmetric *ν*C–H stretching peaks at 2935 cm^−1^ and symmetric peaks at 2885  cm^−1^ in *S*. *salivarius* K12 and *L*. *mesenteroides* DSM 20343 correspond to the methylene group CH_2_ [47,48]. A broad *ν*C–H stretching absorption band with an asymmetric peak at 2927.94 cm^−1^ was distinguished in the DEX spectrum [49,50]. The stretching vibration of the existence of *ν*C–H and the broad band at 1647 cm^−1^ in all EPSs analyzed specify residual water presence [48,51]. The appearance of a *δ*C–O–H bending vibration at 1380–1340 cm^−1^ was also observed and could serve as a feature of the fructose oligomers and polymers [52].

The bands at 1458 cm^−1^ and 1410 cm^−1^ in the BL produced by *S*. *salivarius* K12 and *L*. *mesenteroides* DSM 20343 represent bending vibrations of CH_2_ and CH_3_, respectively [53,54,55]. Broad and intense absorption bands due to C–O stretching vibrations were observed at 1126.43 cm^−1^ and 1014.56 cm^−1^, indicating the presence of pyranose or furanose units in the produced EPSs with C–OH or C–O–C vibrations, which are a carbohydrate fingerprint [44]. Characteristic absorption at 923, 873, and 809 cm^−1^ in the BL of *S*. *salivarius* K12 and *L*. *mesenteroides* DSM 20343 was also observed, corresponding to the furanose ring of the sugar units [44,56]. The acquired spectra confirm the molecular fingerprint attributed to the basic skeleton and functional groups of EPSs, particularly BL. Both EPSs shared similar structures. The values are consistent with those of commercial BL reported by [57,58,59].

### 3.2. Surface Morphology and Microstructural Feature Analysis of Bacterial Levan Synthesized by Streptococcus salivarius K12 and Leuconostoc mesenteroides DSM 20343 Using Scanning Electron Microscopy

According to an observation made by Qin et al. [60], a microstructure serves as an indicator allowing for the prediction of physical properties of unknown polymers and defining their further applicability within the industry. For instance, the highly branched and microporous structure of BLs and the availability of hydroxyl groups explain its ability to form rigid hydrogels with multiple applicability within the food and pharmaceutical industries [61,62,63]. The surface morphology and microstructure of freeze-dried and ground BL produced by *S. salivarius* K12 (B) and *L. mesenteroides* DSM 20343 were acquired by taking advantage of SEM and are depicted in Figure 4. Both BLs show a crystalline morphology with a smooth and glossy surface at the splitting site, forming sharp edges. Similar surface features were reported by Shimizu et al. [64] and Pei et al. [65] for BLs isolated from *Erwinia herbicola* and *Bacillus megaterium* PFY-147, respectively. A cross-section of specimens revealed a poroid, concrete-like network structure with less than 3 μm diameter of pores within matrices of BL isolated from *S. salivarius* K12 (B) and *L. mesenteroides* DSM 20343. However, the former BL demonstrated the most uniform pore distribution. The observed microstructure of BLs does not resemble morphological features of high-molecular-weight BL isolated from *S*. *salivarius*, as documented by Xu et al. [66], presumably due to differences in biosynthesis, extraction, and purification conditions. The observed microstructural features, as well as the available literature, allow for the speculation that BLs produced by two LAB species would find their applicability within the food, pharmaceutical, and cosmetic industries. However, additional experiments elucidating structural–mechanical analysis are necessary.

### 3.3. Characterization of Bacterial Levan Biosynthesized by Streptococcus salivarius K12 and Leuconostoc mesenteroides DSM 20343 by Multi-Detector Gel Permeation Chromatography/Size-Exclusion Chromatography–Liquid Chromatography

The molecular structure and physical properties of BL biosynthesized by *S. salivarius* K12 and *Leuconostoc mesenteroides* DSM 20343 were determined by the GPC/SEC–HPLC system. The multi-detection chromatograms of the BL indicate that good signal-to-noise ratios were obtained, and therefore, absolute molecular weight and intrinsic viscosity calculations could be estimated (Figure 5A,B).

The RI detector in red shows the sample’s concentration and thus reveals that the two samples eluted at different times. Following the RI detector, the BL biosynthesized by *L*. *mesenteroides* DSM 20343 eluted much later (17–21 mL) than that biosynthesized by *S. salivarius* K12 (14–20), indicating that it is much smaller in M_w_. The right-angle (RALS) and left-angle (LALS) scattering detectors show that the BL biosynthesized by *L. mesenteroides* also had a peak around 14–16 mL, but this was in such a low concentration that the RI detector did not detect it. This phenomenon is typical for particles with an extremely high molecular weight; following the Rayleigh–Gans–Debye theory of static light scattering, they scatter light proportionately to their molecular weight and change their solution refractive index proportionally to the sample’s concentration [67]. The appearance of an additional peak may have indicated the presence of a component of lower molecular weight in the biopolymer fraction. Santos-Moriano et al. [68] made a reasonably similar observation, highlighting that the production of BL by the levansucrase of *Zymomonas mobilis* at lower operational temperatures contributes to the formation of longer polysaccharides. Figure 5C,D show that the duplicate injections of the samples overlayed well, indicating that the samples were stable in-solution. A lack of intermolecular interactions with the column packing material and the filter materials illustrates that the chosen conditions for SEC were suitable. The numerical values of BL are summarized in Table 3, indicating that BL biosynthesized using *S. salivarius* K12 had an M_w_ 500 times larger than BL obtained by *L. mesenteroides* DSM 20343, whose values corresponded to 15.435 × 10^4^ and 26.6 kDa, respectively. The obtained data reveal that BL produced by *L. mesenteroides* DSM 20343 is classified as a low-molecular-weight polymer, according to Hou et al. [69]. However, this is the first report demonstrating the ability of *S. salivarius* K12 to produce BL over 15 million Da under standardized processing and operational conditions. The observed M_w_ of BL of *S. salivarius* K12 is in agreement with that reported by Newbrun and Baker [70] for BL derived from the reference test culture *S. salivarius* ATCC 13419, corresponding to 16–23 million Da. This degree of polymerization of levan makes it possible to use it as both a capping and reducing agent in the green biosynthesis of metallic NPs due to its intrinsic immunomodulatory activity and its physical–chemical properties [63]. Its unique features such as water solubility, strong adhesiveness, film-forming ability, low viscosity, high oil solubility, compatibility with salts and surfactants, thermo-, acid and alkaline resistance, and high water retention capacity make BL useful in the preparation of hydrogels and CoSs containing metallic NPs suitable for wound dressing applications [71]. Polydispersity estimation (PD = M_w_/M_n_) of the produced BL displayed a monodispersed molecule isolated from *S*. *salivarius* K12, corresponding to 1.08 and 2.17 for the *L*. *mesenteroides* DSM 20343 isolate (Table 3). The estimated PD value of BL derived from *S*. *salivarius* K12 is consistent with that of Ua-Arak et al. [72] for BL isolated from *Gluconobacter albidus* TMW 2.1191, revealing narrow molar mass distribution among molecules within BL. However, the polydispersity value of BL produced by *L*. *mesenteroides* DSM 20343 was relatively higher than that of BL from *S*. *salivarius* K12. The presence of tails on chromatograms presumably suggests the branching structure of BL produced by *L*. *mesenteroides* DSM 20343 [72]. The hydrodynamic radius (Rh), which reflects the shape of molecules, estimated for the BLs of *S*. *salivarius* K12 and *L*. *mesenteroides* DSM 20343, was found to be, on average, 35.7 and 3.5 nm, respectively.

The observed R_h_ value of BLs produced by *S*. *salivarius* K12 aligns with that reported by Zhang et al. [73] for EPS derived from yoghurt fermented with a high-EPS-producing strain *S. thermophilus* AR333. The unusually low intrinsic viscosity of BL produced by *S*. *salivarius* K12 and *L*. *mesenteroides* DSM 20343 revealed a compact spherical molecular structure [74]. The observed values are consistent with those reported by Semjonovs et al. [30] and Arvidson et al. [75]. The recovery of BLs produced, which was estimated between the injected and detected concentrations of BL, corresponded to 54.7 and 81.2%. A reduced recovery for the BL produced by *S. salivarius* K12 was conditioned by a relatively larger M_w_ of the EPS molecule and difficulties associated with complete dissolution by selected solvents used for sample preparation during GPC/SEC–HPLC analysis.

### 3.4. Compositional Analysis of Bacterial Levan Produced by Streptococcus salivarius K12 and Leuconostoc mesenteroides DSM 20343

According to the available literature, BL is a homopolysaccharide of fructose produced extracellularly by a variety of bacterial species [76]. The structure of BL is well characterized, indicating the presence of D-fructofuranosyl backbone residues linked by β-(2→6) glycosidic linkages at the C-1 position of the fructose ring [30,31]. Moreover, β-(2→6) linkages make up 70% of the total linkages, while the remaining 30% represent β-(2→1) branches [77]. Since the initial molecule in BL formation is sucrose, either internal or terminal glucose moiety can be aperient as a by-product of BL biosynthesis by levansucrase (EC 2.4.1.10), responsible for the catalysis of sucrose [78]. A detailed profile of individual carbohydrates of BL is depicted in Figure 6 and Table 4.

The presence of sucrose, glucose, fructose, galactose, and lactose as free carbohydrate representatives was confirmed chromatographically in both produced BLs. Sucrose and galactose were the main sugars found in the BL derived from *S. salivarius* K12 and *L. mesenteroides* DSM 20343, corresponding to 1.2 and 2.5 g 100 g^−1^, respectively. The presence of the above free/soluble sugars was conditioned by the composition of the nutrient medium used for growing bacteria. The observed values revealed that the principal component of produced BLs is fructose, the concentration of which varied in range from 56.9 to 92.0 g 100 g^−1^ and from 38.4 to 62.3 g 100 g^−1^ in *S. salivarius* K12 and *L. mesenteroides* DSM 20343, respectively. As seen, hydrolysis of BL with oxalic acid in both cases was found to be the most efficient, as 100 g of BL delivered roughly 92% and 62.3% of fructose, respectively. This observation can be supported by an early study by Murphy [79], indicating 100% fructose recovery after 9 h of BL hydrolysis with 0.1 M oxalic acid at 100 °C. A relatively lower but still relevant fructose yield was obtained by applying aqueous HCl solutions of various molarities. The highest yield of fructose was reached after 3 h of hydrolysis with 0.1 M HCl at 90 °C in the case of BL from *S. salivarius* K12. In comparison, a higher degree of hydrolysis for *L. mesenteroides* DSM 20343 was found after 6 h. It was observed that with increasing molarity of the HCl solution, there was a noticeable decrease in the fructose content, possibly due to fructose dehydration to hydroxymethylfurfural [80]. Moreover, the non-selectivity of HCl as a catalyst in cleavage of glycosidic linkages and weak fructose release from BL produced by *Zymomonas mobilis* was highlighted by Bekers et al. [81] and Kennedy et al. [82] as a high amount of fructooligosaccharides was observed in the final products. The same trend of increase as a function of HCl molarity was observed for acid-treated BL produced by *S. salivarius* K12. As seen, the highest glucose yield was reached after hydrolysis of BL with 0.5 M HCl for 6 h, corresponding to 5.8 g 100 g^−1^. Generally, the ratio of fructose to glucose in BL of *S. salivarius* K12 was found to be 94.8:4.3, which is reasonably close to the ratio reported by Matulová et al. [83] for BL produced by *Bacillus* sp. 3B6. It is worth noting that the composition of BL produced by *L. mesenteroides* along with fructose was represented by the relatively high amount of galactose rather than glucose. The exact mechanism of galactose formation remains unclear. However, the contribution of both α- and β-galactosidases of *L. mesenteroides* DSM 20343 in the hydrolysis of lactose, which originated from the nutrient medium, may have explained the abundancy of galactose, and only a trace level of lactose detected in the final hydrolysates of BL [84].

### 3.5. Spectrophotometric Analysis of Elaborated Colloid Systems Based on Bacterial Levan and Silver Nanoparticles by UV–Vis Spectroscopy

A preliminary experiment for confirming AgNP formation during green biosynthesis using BL as a reducing and capping agent at different concentrations was monitored by UV–Vis absorption spectra. Under alkaline circumstances, reached by adding 50.0 µL 1.0 M NaOH to the working solution, a gradual color change from light yellow to reddish brown and dark blue occurred during thermal exposure for 1 h, revealing the formation of BL@AgNPs (Figure 7). The absorption band near the red shift at 400 nm is surface plasmon resonance (SPR) of AgNPs, as reported by Kemala et al. [85]. The most intense peak appeared at 416 and 424 nm using 20 mg BL from *S*. *salivarius* K12 and *L*. *mesenteroides* DSM 20343, respectively (Figure 7A,B). However, as the concentration of BL increased, the formation of a band at 294 nm began to appear, which reached its maximum intensity when 500 mg BL was applied. The band at 294 may correspond to the absorption of a raw component, the BL itself, used in the biosynthesis of AgNPs [86]. It has been observed that with the increase in the band at 294 nm as a function of BL concentration, a substantial decrease of the main band at 400 nm became apparent. The observation can be attributed to the lower conversion of silver ions Ag^+^ to AgNPs due to the process termination caused by EPSs and a possible change in size/shape and/or concentration of BL@AgNPs in the medium after an increase in EPSs in the biosynthesis mixture [87]. A credible explanation for this multilayer adsorption was provided by Jeong et al. [88], attributing this phenomenon to the photoluminescence of photoexcited material when it starts to release its energy as light.

This statement allows for the speculation of the possible formation of a rigid, dense coating of the EPS molecules in the AgNP surface and the changing of Ag nanostructures from NPs to nanofilms due to the increasing thickness of Ag in the CoS. This is reinforced by an observation made by Rosa et al. [89] and also by Varghese et al. [90], highlighting the cross-linking of polyvinylpyrrolidone (PVP) nanofibers and high-molecular-weight compounds (proteins in particular) present in plant extracts and AgNPs, respectively, and the formation of a new resonance peak at 290 nm in the UV–Vis spectrum from the Ag shell after the Ag shell reached a certain thickness. However, in the case of commercial high-molecular-weight DEX, neither 50 mg nor 500 mg of raw material contributed to forming a band at 290 nm, indicating the presence of solely DEX-capped AgNPs in the developed CoS. Indeed, the most intense, sharp, and symmetric band appeared at 403 nm using 500 mg of DEX as a reducing and capping agent. This observation allows for the assumption of forming more uniform in shape and size AgNPs, as conveyed by [41,91].

### 3.6. Structural Analysis of Bacterial Levan-Capped Silver Nanoparticles

The examination of biosynthesized BL@AgNP microstructures established NP formation, as shown in Figure 8. The micrographs revealed the availability mainly of monodispersed and uniformly distributed sphere-shaped BL@AgNPs in CoS engineered through the utilization of 20 (S_20 mg, L_20 mg) and 30 mg (S_30 mg, L_30 mg) of BL from *S*. *salivarius* K12 and *L*. *mesenteroides* DSM 20343, respectively. However, an insignificant part of agglomerating BL@AgNPs in these CoSs was observed due to the “depletion flocculation” phenomenon, as highlighted by Saravanan et al. [92], caused by the elevated concentration of PVP in the solution. According to measurements completed by SEM, the average sizes of the BL@AgNPs corresponded to 12.67 ± 5.56 and 12.58 ± 4.58 nm for NPs biosynthesized by 20 and 30 mg of BL from *S*. *salivarius* K12 and to 14.96 ± 10.25 and 15.16 ± 6.15 nm for NPs biosynthesized by 20 and 30 mg of BL from *L*. *mesenteroides* DSM 20343, respectively. As the BL concentration in the reaction mixture increased, a substantial increase in NP size was noted in the case of BL from *S*. *salivarius* K12, while it did not exceed 46.97 ± 20.23 nm, even though it had a broad distribution. As revealed by González-Garcinuño et al. [32], the size of BL-capped AgNPs was 36.9 ± 11.8 nm using 20 mg of BL as a reducing and stabilizing agent, and no apparent aggregation phenomenon was observed due to significant repulsive forces ensured by the negative surface charge of NPs. In the case of AgNPs produced using BL from *L*. *mesenteroides* DSM 20343, the size varied from 13.17 ± 2.76 to 28.11 ± 9.27 nm for 50 mg and 500 mg of a reducing and capping agent, respectively. The presence of additional free functional groups in the branched chains of amphiphilic BL from *L*. *mesenteroides* DSM 20343 due to glucose and galactose residues and better solubility in water due to lower molecular weight perhaps were the main factors that promoted the formation of smaller in size BL@AgNPs with a narrow distribution. Superior distribution and smaller in size AgNPs were biosynthesized by commercial DEX as a reducing and capping agent, which was conditioned by the presence of glucose as the main component of DEX and its readiness rather than fructose to act as a reducing agent after aldehyde group oxidation under alkaline conditions [93]. The average size of DEX@AgNPs corresponded to 24.31 ± 13.88 and 12.99 ± 3.84 nm using 50 and 500 mg of DEX, respectively. The biosynthesis of AgNPs from DEX required a more reducing agent to yield nano-object quantities comparable to 20 mg of *S*. *salivarius* K12 and *L mesenteroides* DSM 20343. Encouraging results were reported by Carré-Rangel et al. [91], highlighting the formation of spherical and more diminutively sized AgNPs using DEX of different molecular weights as a reducing and capping agent. The presence of AgNPs in the CoSs was confirmed by EDX analysis, displaying distinctive signal peaks within the 3.0–3.4 keV range due to SPR [94]. The EDX pattern at four randomly selected regions revealed the presence of Si (silicon) and Na (sodium) elements due to the Si wafer used as a substrate during SEM analysis and NaOH as an accelerator in the formation of AgNPs in the green biosynthesis, respectively (Figure 8A). The contribution of the above elements to the total weight was below 1% of the weight, indicating the relative purity of the obtained AgNPs. The presence of BL in the CoSs manifested in the form of an intense C (carbon) peak, which was found to be the second prevalent element observed along with Ag.

### 3.7. Antimicrobial Activity of Bacterial Levan and Colloid Systems Containing Silver Nanoparticles According to Diameter of Inhibition Zone Values

In the following experiments, the antimicrobial activity of obtained CoSs containing AgNPs was assessed using either disc or well diffusion approaches. The selection of two antimicrobial activity evaluation methods was based on evidence regarding the lack of direct interconnection of NPs deposited on the disc with bacteria distributed over the media. Due to the relatively lower diffusivity magnitude of NPs, the Kirby–Bauer method gave only a vague representation of the antimicrobial activity of AgNPs and no direct comparison with topical antibiotics could be made [39]. Therefore, for comparative purposes, antimicrobial activity probing was conducted by the direct contact of AgNPs with microorganisms, taking advantage of agar well diffusion approach. The disc and well diffusion test results show substantially different inhibition zones between the two antimicrobial probing methods (Figure 9 and Table 5). As seen, the most remarkable inhibition of selected bacteria was observed by the well diffusion test. The obtained results support the finding made by Bubonja-Šonje et al. [95], indicating that agar well diffusion is a more sensitive and convenient approach than that of disc diffusion, especially suitable for cationic compounds or their mixtures. The pronounced zone of inhibition was highlighted for the CoS containing AgNPs biosynthesized by 500 mg BL BLss@AgNPs 500, as the growth of the selected bacteria was inhibited the most. The most susceptible bacterium to elaborated CoS was found to be negatively charged *P. aeruginosa* ATCC 27853, followed by *E*. *coli* ATCC 25922; their zones of inhibition corresponded to 21.2 and 19.3 mm, respectively. It is worth noting that these observed inhibition zones are more extensive than those reported by [96,97,98]. However, based on the results from both diffusion tests, it turned out that the elaborated CoSs, regardless of the size of AgNPs they contained, were found to be less effective against clinical isolates with MDR, for *P*. *aeruginosa* in particular. The resistance of *P*. *aeruginosa* is conditioned by its ability to produce a biofilm that protects it from environmental threats, as reported by Maurice et al. [99]. The observed data are reinforced by El Din et al. [100], revealing a low susceptibility of the clinical MDR isolate *P. aeruginosa* to the AgNPs with calculated MIC and MBC values as high as 8.0 mg mL^−1^. The elaborated CoS with 500 mg of BL BLss@AgNPs 500 showed exceptional inhibitory activity against clinical isolates with MDR, i.e., *S. epidermidis*, *E. coli*, and *E. faecium*, corresponding to 16.2, 14.4, and 14.3 mm zones of inhibition, respectively. The observed values are consistent with those reported by Abeer et al. [101] for AgNP conjugates with such antibiotics as ciprofloxacin, cefotaxime, and cefazolin. It is worth noting that exactly the BL BLss@AgNPs 500 CoS with an average of 46.97 ± 20.23 nm NPs which showed the presence of a new resonance peak at 290 nm in the UV–Vis spectrum and hypothetically belongs to the conjugates of AgNPs and BL in the form of a rigid nanofilm, demonstrated antimicrobial activity the most. The elaborated matrix allowed for the extension of the release of Ag^+^ ions during the entire incubation period, thus extending the antimicrobial effect. The elaborated CoS with 50 mg of BL BLss@AgNPs 50 with a size of NPs 16.37 ± 3.91 nm ensured almost equal antimicrobial contribution to suppressing the growth of the selected reference bacteria and clinical isolate. As for the reference test cultures, the zone of inhibition varied from 12.4 to 17.4 mm, while for clinical isolates, from 11.3 to 15.1 mm. The most susceptible bacterium to BLss@AgNPs 50, as in the case of BLss@AgNPs 500, was *P. aeruginosa* ATCC 27853, followed by *E*. *coli* ATCC 25922 and the clinical isolate *S. epidermidis*. Considering this circumstance, the designed CoSs as nanocomposites with the sustained release of Ag^+^ ions and long-lasting bactericidal effects may find applicability in wound management. The proposed nanocomposite as a non-toxic and cost-effective API is of potential interest as an alternative to other antiseptics and topical antibiotics that can be incorporated into hydrogels and used locally [102].

The MIC and MBC values were estimated to complement the abovementioned tests (Table 6). The prominent antimicrobial activity of elaborated CoS-BL@AgNPs against selected reference test cultures was noticed for those mediated by either 20 or 50 mg of BL derived from *S. salivarius* K12 and *Leuconostoc mesenteroides* DSM 20343. The MIC values of CoS-BL@AgNPs introduced to the wells varied in a range from 2.6 to 20.8 µL mL^−1^, respectively. The most substantial contribution of CoSs to the inhibition of bacteria was noticed for AgNPs, with their size ranging from 12.58 to 16.37 nm, thus reinforcing the importance of NP size during direct interaction with cell membranes [103]. *P. aeruginosa* ATCC 27853 was the most susceptible to elaborated CoS. The values of MIC and MBC were found as follows: 2.6 and 5.2 µL mL^−1^ for BLss@AgNPs 20, respectively. At the same time, the other CoSs were found to be equally effective against this bacterial species. Outstanding antimicrobial activity of BLss@AgNPs 50 was highlighted against *E. coli* ATCC 25922, with MIC and MBC values corresponding to 2.6 µL mL^−1^ of elaborated CoS. However, it took 10.4 µL mL^−1^ of CoS-BL@AgNPs mediated by 20 mg BL to induce the death of this bacterial species. From the list of reference test cultures selected, *S*. *aureus* ATCC 6538P was found to be the most resistant bacterium as 20.8 µL mL^−1^ of elaborated CoSs needed to suppress the growth of this bacterium and 41.7 µL mL^−1^ to cause irreversible consequences that resulted in death. The observed MIC value is consistent with those reported by Ansar et al. [104] and Ashour et al. [105] for AgNPs mediated by *Brassica oleracea*- and *Vaccinium macrocarpon*-derived extracts, respectively. The relatively better resistance of *S*. *aureus* ATCC 6538P to CoSs is conditioned by its superior film-forming ability and antibiotic-resistant spontaneous generation of small-colony variants (SCVs) [106].

Similar to the reference test culture, the clinical isolate of *P. aeruginosa* exhibited relatively lower resistance to CoS, as only 5.2 µL mL^−1^ of elaborated CoSs was necessary to cause irreversible consequences resulting in death. The observed values required to inhibit the growth of MDR *P. aeruginosa* were found to be substantially lower than those reported by [107,108]. Interestingly, *S. epidermidis*, which is described as a biofilm-producing MDR [109], was found to be less resistant to elaborated CoS than that of the reference test culture, as only 20.8 µL mL^−1^ of elaborated CoS was found to be the lower concentration that led to the death of this bacterial species. It follows that timely treatment with elaborated CoS can prevent the development of biofilm-producing MDR pathogen, thereby contributing to the reduction of inflammatory processes. Swolana et al. [110] observed almost equal effectiveness of engineered AgNPs in suppressing *S. epidermidis* with biofilm-forming ability. The effectiveness of engineered CoS with a size of AgNPs ranging from 12.58 to 16.37 nm in relation to *E. coli* inhibition should be pointed out, as only 5.0 µL mL^−1^ was necessary for inhibition and complete death of this clinical isolate with MDR. The remarkable antimicrobial potency of commercially available AgNPs has been reported by Dove et. al. [111], revealing the ability of pure AgNPs in the amount of 3.1 µL mL^−1^ and AgNPs in conjunction with ribosome-targeting antibiotics such as aminoglycosides to cause defects in the electron transport chain of *E*. *coli* with the new resistance gene mcr-1. Overall, the elaborated CoS-BL@AgNPs can be treated as APIs suitable for designing new antimicrobial agents and modifying therapies in controlling multi-drug-resistant (MDR) pathogens.

## 4. Conclusions

Due to the narrow availability of evidence on the elaboration of CoS-AgNPs using the biotechnologically produced exopolysaccharide levan as a reducing agent, and given the need to develop antimicrobials with long-lasting effects against bacteria with multi-drug resistance and biocompatibility with the human body, this study demonstrated a potential utilization of bacterial levan produced by two species of lactic acid bacteria, i.e., *Streptococcus salivarius* K12 and *Leuconostoc mesenteroides* DSM 20343, as non-toxic reducing agents. The molecular fingerprinting performed by multi-detector size-exclusion chromatography revealed that the M_w_ of BL biosynthesized by *S. salivarius* K12 made up 15.435 × 10^4^ kDa, making it attainable to use as a capping agent and hydrogel-based wound dressing. In producing CoS-BL@AgNPs following green biosynthesis, the presence of nanostructured objects with size distributions from 12.67 ± 5.56 nm to 46.97 ± 20.23 nm was confirmed by SEM. Due to the cationic nature of the elaborated CoSs, two independent diffusion tests displayed substantially different inhibition zones. The well diffusion test underlined the most remarkable inhibition of reference test cultures and those of clinical origin. The foremost inhibitory potency of the designed CoSs was reinforced by minimum inhibitory and bactericidal concentrations, demonstrating its conceivable utilization as an active pharmaceutical ingredient for producing antimicrobials that could effectively contribute to controlling bacteria with multi-drug resistance.

## Figures and Tables

**Figure 1 nanomaterials-13-02969-f001:**
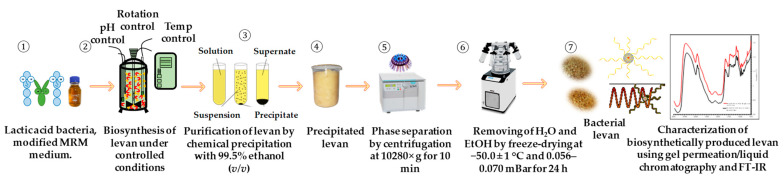
A schematic representation of major steps undertaken to prepare bacterial levan by two lactic acid bacteria species, i.e., *Streptococcus salivarius* K12 and *Leuconostoc mesenteroides* DSM 20343 with subsequent purification by chemical precipitation.

**Figure 2 nanomaterials-13-02969-f002:**
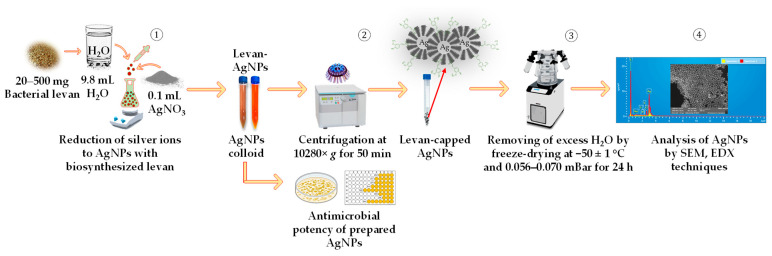
A schematic representation of major steps undertaken to prepare silver nanoparticle colloids and purified nanoparticles following the green synthesis principle using bacterial levan recovered from *Streptococcus salivarius* K12 and *Leuconostoc mesenteroides* DSM 20343.

**Figure 3 nanomaterials-13-02969-f003:**
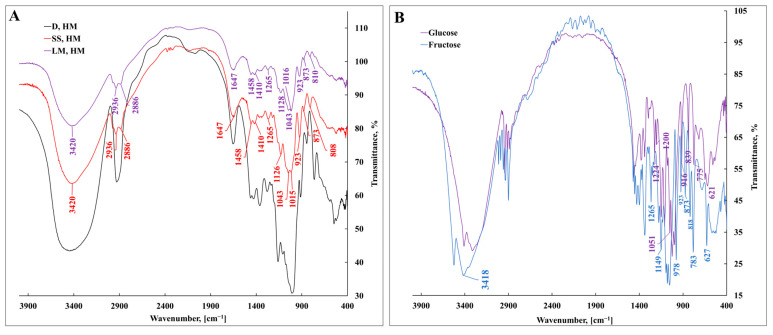
Comparison of the FT-IR spectrum for produced bacterial levan by two species of lactic acid bacteria: (**A**) *Streptococcus salivarius* K12—SS, HM and *Leuconostoc mesenteroides* DSM 20343—LM, HH, and standard dextran (control) from *Leuconostoc mesenteroides*—D, HM. The vibrational FT-IR spectrum is compared with α-D-glucose and β-D-fructose commercial standards (**B**).

**Figure 4 nanomaterials-13-02969-f004:**
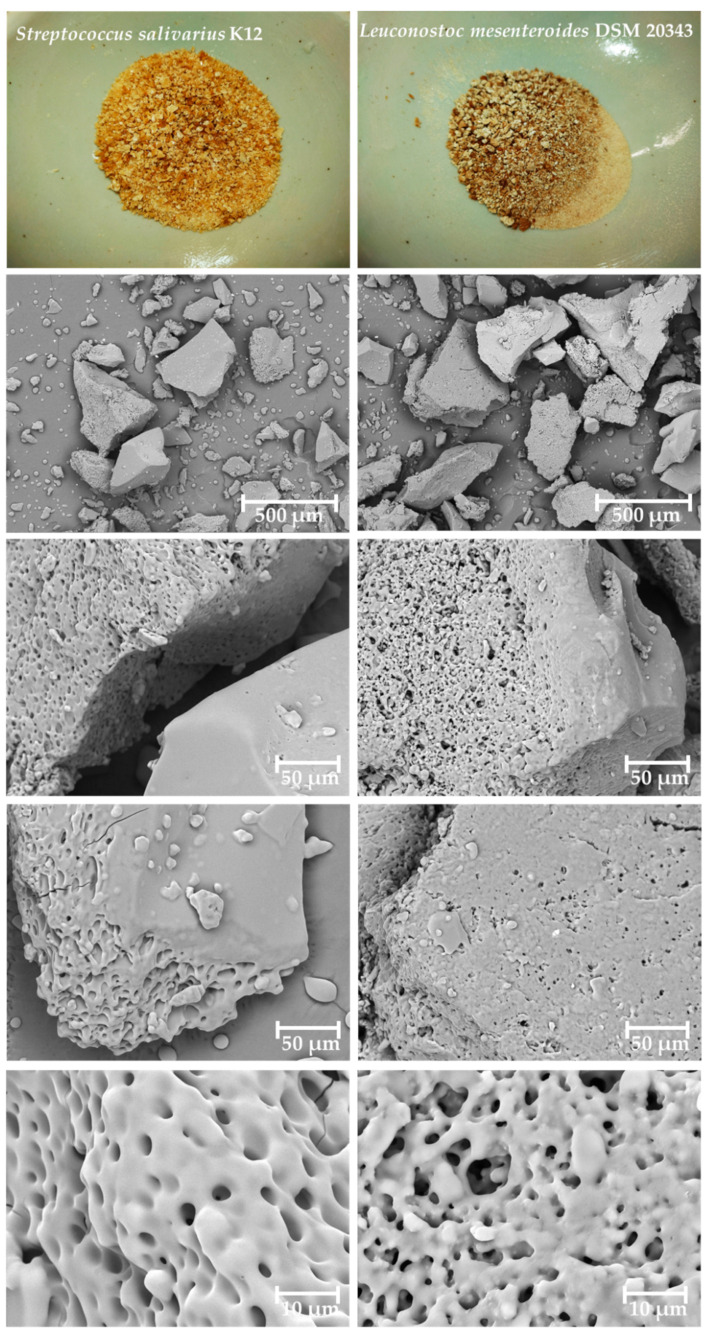
Representative scanning electron microscope micrographs of bacterial levan biosynthesized by *Streptococcus salivarius* K12 (B) and *Leuconostoc mesenteroides* DSM 20343 acquired under 150 x, 1.0 kx, and 5.0 kx times magnification.

**Figure 5 nanomaterials-13-02969-f005:**
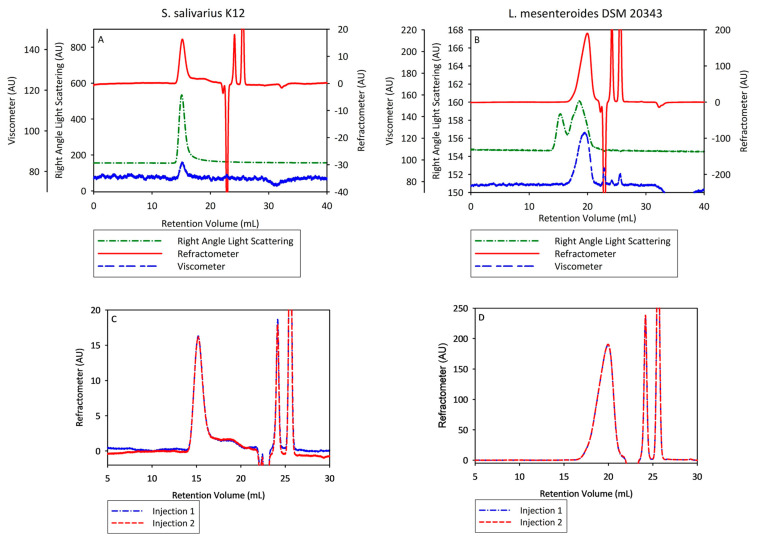
Multi-detector size-exclusion chromatograms of bacterial levan isolated from *Streptococcus salivarius* K12 (**A**) and *Leuconostoc mesenteroides* DSM 20343 (**B**) with 100 µL injection at concentrations of 6.2 and 6.6 mg mL^−1^, respectively. The overlay of the refractive index chromatograms for the bacterial levan isolated from *Streptococcus salivarius* K12 (**C**) and *Leuconostoc mesenteroides* DSM 20343 (**D**) samples, respectively, shows a relatively small deviation between permeation peaks of 2 injections during detection by refractive index. ***Note:*** refractive index (red); right-angle light scattering (green); low-angle light scattering (black); viscometer (blue).

**Figure 6 nanomaterials-13-02969-f006:**
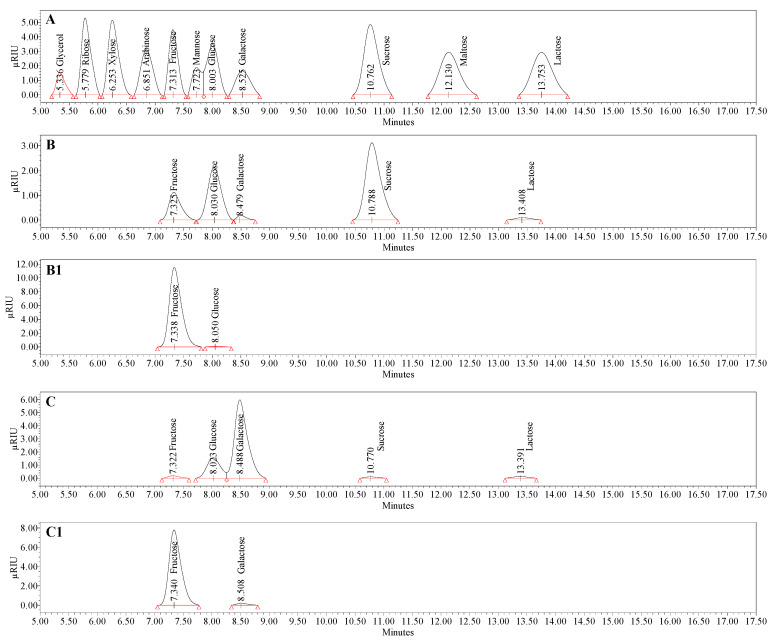
HPLC-RID chromatogram representing the separation of carbohydrates of a standard mixture (**A**), bacterial levan biosynthesized by *Streptococcus salivarius* K12 (**B**), and *Leuconostoc mesenteroides* DSM 20343 (**C**). ***Note:*** The profiles of saccharides in the (**B1**) and (**C1**) chromatograms refer to released monomers recovered after complete hydrolysis of bacterial levan biosynthesized by *Streptococcus salivarius* K12 and *Leuconostoc mesenteroides* DSM 20343 with 0.05 M oxalic acid for 3 h at 90 °C, respectively.

**Figure 7 nanomaterials-13-02969-f007:**
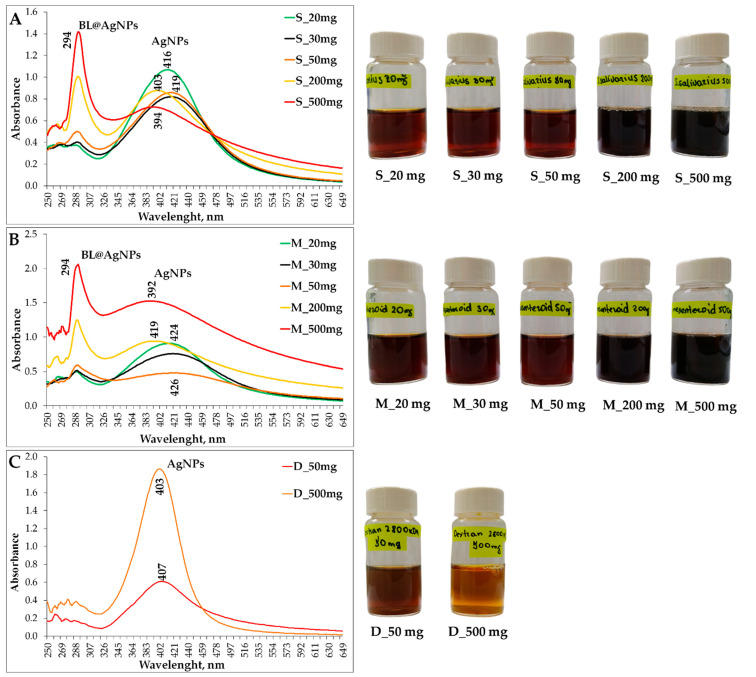
UV–Vis spectrum of colloid systems containing silver nanoparticles reduced with different concentrations of bacterial levan biosynthesized by *Streptococcus salivarius* K12 (**A**), *Leuconostoc mesenteroides* DSM 20343 (**B**), and high-molecular-weight dextran as the control (**C**), representing absorption peaks within a range of 392–426 nm. Representative color change in colloid systems as a function of reduction agent concentration applied during green biosynthesis of silver nanoparticles. ***Note:*** Capital letters S and M refer to bacterial levan used for the biosynthesis of AgNPs, i.e., *Streptococcus salivarius* K12 and *Leuconostoc mesenteroides* DSM 20343, respectively. The capital letter **D** refers to the commercial high-molecular-weight dextran used for the biosynthesis of AgNPs. The numbers indicate the amount of exopolysaccharide used as a reducing and capping agent.

**Figure 8 nanomaterials-13-02969-f008:**
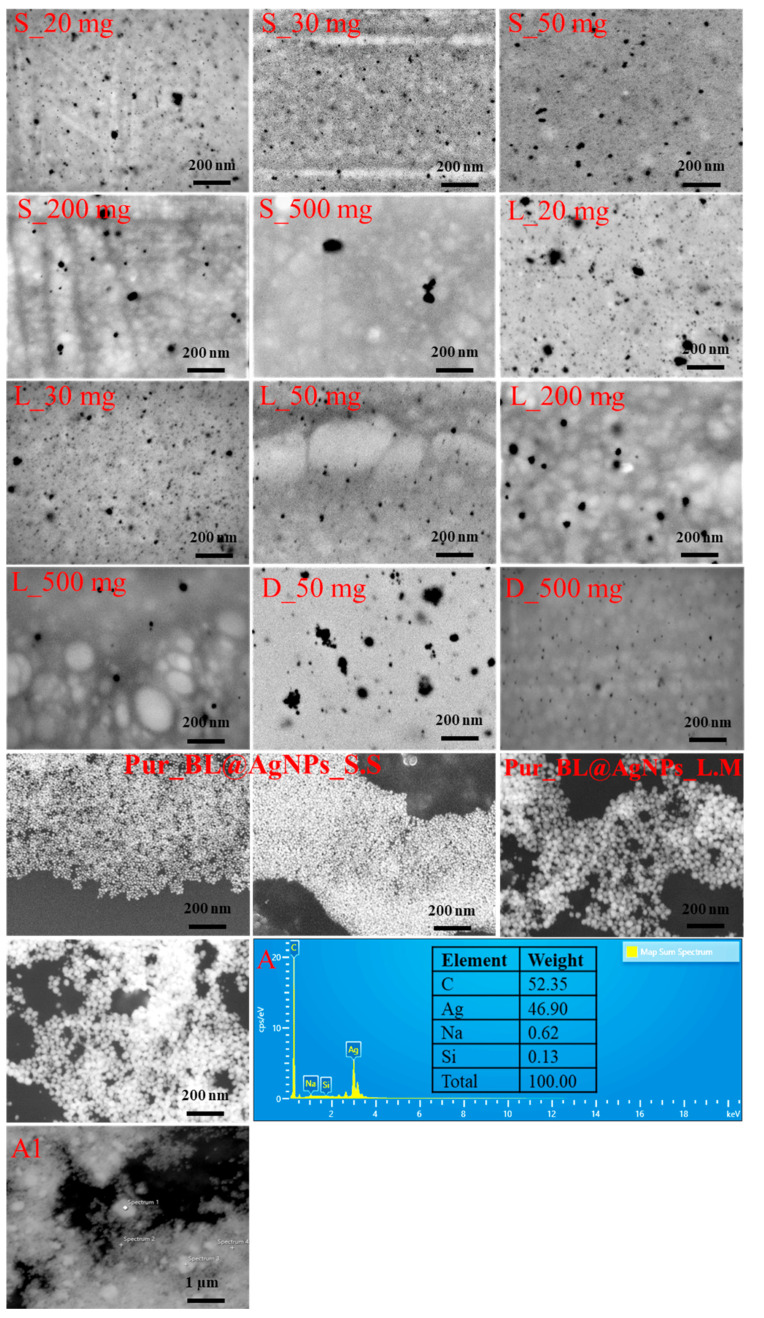
Scanning electron microscopy (SEM) images of biosynthesized bacterial levan-capped silver nanoparticles in colloid systems produced by green biosynthesis. Energy-dispersive X-ray spectroscopy (EDX) pattern (**A**) of silver nanoparticles produced by bacterial levan captured in four randomly selected regions (**A1**) for EDX spectroscopy analysis. ***Note:*** SEM micrographs: capital letters S and L refer to the bacterial levan used for the biosynthesis of AgNPs, i.e., *Streptococcus salivarius* K12 and *Leuconostoc mesenteroides* DSM 20343, respectively. The capital letter **D** refers to the commercial high-molecular-weight dextran used for the biosynthesis of AgNPs. The numbers indicate the amount of exopolysaccharide used as a reducing and capping agent. Pur-BL@AgNPs_S.s and Pur-BL@AgNPs_L.M represent monolayers of bacterial levan-capped silver nanoparticles after purification.

**Figure 9 nanomaterials-13-02969-f009:**
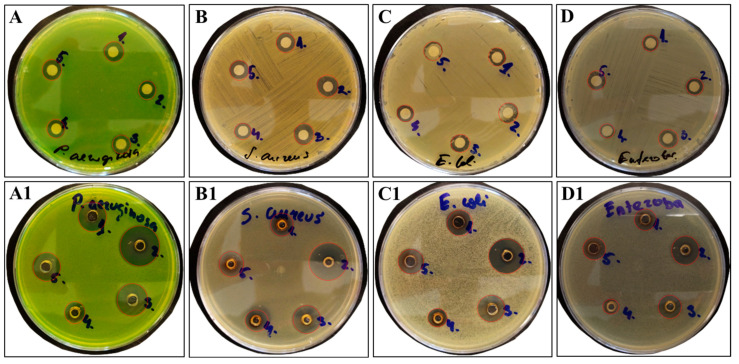
Disc (upper row) and well diffusion (lower row) tests for colloid systems contained silver nanoparticles produced within green biosynthesis by bacterial levan derived from *Streptococcus salivarius* K12 and *Leuconostoc mesenteroides* DSM 20343 and commercial high-molecular-weight dextran against *Pseudomonas aeruginosa* ATCC 27853 (**A**,**A1**), *Staphylococcus aureus* ATCC 6538P (**B**,**B1**), *Escherichia coli* ATCC 25922 (**C**,**C1**), and *Enterococcus aerogenes* ATCC 13048 (**D**,**D1**). ***Note:*** 1—colloid system containing silver nanoparticles biosynthesized using 20 mg of bacterial levan from *Streptococcus salivarius* K12 (BLss@AgNPs 20); 2—colloid system containing silver nanoparticles biosynthesized using 500 mg of bacterial levan from *Streptococcus salivarius* K12 (BLss@AgNPs 500); 3—colloid system containing silver nanoparticles biosynthesized using 50 mg of bacterial levan from *Streptococcus salivarius* K12 (BLss@AgNPs 50); 4—colloid system containing silver nanoparticles biosynthesized using 500 mg of high-molecular-weight dextran (Sigma-Aldrich) (BLdex@AgNPs 500); 5—colloid system containing silver nanoparticles biosynthesized using 20 mg of bacterial levan from *Leuconostoc mesenteroides* DSM 20343 (BLlm@AgNPs 20).

**Table 1 nanomaterials-13-02969-t001:** The composition of the De Man, Rogosa, and Sharpe modified broth used for the biosynthesis of bacterial levan by lactic acid bacteria *Streptococcus salivarius* K12 and *Leuconostoc mesenteroides* DSM 20343.

Component	Chemical Formula	Amount, g L^−1^
Bacteriological peptone	–	10.0
Beef extract	–	10.0
Yeast extract	–	5.0
Sucrose	C_12_H_22_O_11_	50.0
Dipotassium hydrogen phosphate	K_2_HPO_4_	2.0
Sodium acetate	C_2_H_3_NaO_2_	5.0
Diammonium citrate	C_6_H_14_N_2_O_7_	2.0
Magnesium sulfate	MgSO_4_	0.2
Manganous sulfate	MnSO_4_	0.05
Tween^®^ 80	C_64_H_124_O_26_	1.0

**Table 2 nanomaterials-13-02969-t002:** FT-IR band assignments for bacterial levan produced by *Streptococcus salivarius* K12 and *Leuconostoc mesenteroides* DSM 20343.

Wavenumber, cm^−1^	Assignment
**IR**	
3524–3314	*ν*OH
3015–2870	*ν*CH symmetric/asymmetric
1647	*ν*CH
1637	*δ*C=O
1458	*δ*CH_2_ + *δ*OCH + *δ*CCH
1410	*δ*CH_3_
1380–1430	*δ*OCH + *δ*COH + *δ*CCH
1340	*δ*CCH + *δ*OCH
1265	Fructose *δ*OH in plane, *δ*CCO
1224, 1200	Glucose *δ*CH + *δ*OH in plane
1149–995	*ν*CO + *ν*CC + *δ*CCC
977	*ν*CO + *δ*CCO
923–916	*ν*CO + *ν*CCH + *ν*_s_ ring of pyranose
873	*ν*CC + *δ*CCH + *δ*CH fructose
818–839	δCH
783,775	*δ*CCH + *δ*CCO
627	CH_2_ + CH
621	CH_2_

***Note:*** *ν*—stretching; as—asymmetric; s—symmetric; β—in-plane bending.

**Table 3 nanomaterials-13-02969-t003:** Detailed description of exopolysaccharide bacterial levan biosynthesized by *Streptococcus salivarius* K12 and *Leuconostoc mesenteroides* DSM 20343.

Replicate	M_w_, g moL^−1^	M_n_, g moL^−1^	PD	R_h_, nm	IV, dL g^−1^	Recovery, %
*Bacterial levan biosynthesized by Streptococcus salivarius* K12						
1	15.146000	14.127000	1.07	36.3	0.21	54.3
2	15.724000	14.508000	1.08	35.3	0.19	55.0
Average	**15.435000**	**14.317500**	**1.08**	**35.8**	**0.20**	**54.7**
STDEV	**0.409**	**0.269**	**0.007**	**0.707**	**0.014**	**0.495**
*Bacterial levan biosynthesized by Leuconostoc mesenteroides* DSM 20343						
1	26.700	12.500	2.14	3.5	0.14	81.8
2	26.400	12.000	2.20	3.5	0.14	80.6
Average	**26.600**	**12.300**	**2.17**	**3.5**	**0.14**	**81.2**
STDEV	**0.212**	**0.354**	**0.042**	**0.000**	**0.000**	**0.849**

***Note:*** M_w_—weight average molecular weight; M_n_—number average molecular weight; PD—polydispersity; R_h_—hydrodynamic radius; IV—intrinsic viscosity; dL—deciliters. Recovery was estimated based on obtained values between the detected concentration and input concentration.

**Table 4 nanomaterials-13-02969-t004:** The concentration of free and bound mono- and disaccharides in bacterial levan produced by *Streptococcus salivarius* K12 and *Leuconostoc mesenteroides* DSM 20343 after hydrolysis for 3 and 6 h and 90 °C with three types of aqueous acid solutions, g 100 g^−1^ DW.

Carb	BL_*S*. *salivarius* K12							BL_*L*. *mesenteroides* DSM 20343						
	Free	Bound, 0.1 M HCl 3 h	Bound, 0.1 M HCl 6 h	Bound, 0.2 M HCl 6 h	Bound, 0.5 M HCl 6 h	Bound, 1.0 M TCA 6 h	Bound, 0.05 M oxalic acid 3 h	Free	Bound, 0.1 M HCl 3 h	Bound, 0.1 M HCl 6 h	Bound, 0.2 M HCl 6 h	Bound, 0.5 M HCl 6 h	Bound, 1.0 M TCA 6 h	Bound, 0.05 M Oxalic Acid 3 h
Gly	n.d.	n.d.	n.d.	n.d.	n.d.	n.d.	n.d.	n.d.	n.d.	n.d.	n.d.	n.d.	n.d.	n.d.
Xyl	n.d.	n.d.	n.d.	n.d.	n.d.	n.d.	n.d.	n.d.	n.d.	n.d.	n.d.	n.d.	n.d.	n.d.
Ara	n.d.	n.d.	n.d.	n.d.	n.d.	n.d.	n.d.	n.d.	n.d.	n.d.	n.d.	n.d.	n.d.	n.d.
Fru	0.4 ± 0.0 ^f^	85.7 ± 0.1 ^b^	84.9 ± 0.1 ^b^	77.1 ± 0.1 ^d^	56.9 ± 0.1 ^e^	80.2 ± 0.1 ^b^	**92.0 ± 0.0 ^a^**	0.1 ± 0.0 ^g^	58.1 ± 0.0 ^c^	60.6 ± 0.00 ^b^	53.1 ± 0.1 ^e^	38.4 ± 0.0 ^f^	55.0 ± 0.0 ^d^	**62.3 ± 0.0 ^a^**
Glu	0.7 ± 0.0 ^e^	2.5 ± 0.0 ^d^	4.7 ± 0.1 ^b^	4.8 ± 0.0 ^b^	5.8 ± 0.0 ^a^	**4.9 ± 0.1 ^b^**	4.2 ± 0.1 ^c^	0.4 ± 0.0 ^c^	BLQ	BLQ	5.3 ± 0.0 ^b^	10.3 ± 0.0 ^a^	5.0 ± 0.1 ^b^	BLQ
Gala	0.3 ± 0.0	n.d.	n.d.	n.d.	n.d.	n.d.	n.d.	2.5 ± 0.0 ^d^	7.3 ± 0.1 ^c^	**12.7 ± 0.1 ^a^**	12.2 ± 0.1 ^ab^	11.9 ± 0.0 ^b^	12.2 ± 0.0 ^a^	12.6 ± 0.0 ^a^
Suc	1.2 ± 0.0	n.d.	n.d.	n.d.	n.d.	n.d.	n.d.	0.05 ± 0.0	n.d.	n.d.	n.d.	n.d.	n.d.	n.d.
Mal	n.d.	n.d.	n.d.	n.d.	n.d.	n.d.	n.d.	n.d.	n.d.	n.d.	n.d.	n.d.	n.d.	n.d.
Lac	0.1 ± 0.0	n.d.	n.d.	n.d.	n.d.	n.d.	n.d.	0.05 ± 0.0	n.d.	n.d.	n.d.	n.d.	n.d.	n.d.
**Tot**	2.7 ± 0.0 ^f^	88.2 ± 0.2 ^b^	89.6 ± 0.2 ^b^	81.9 ± 0.1 ^d^	62.7 ± 0.1 ^e^	85.1 ± 0.2 ^c^	**96.2 ± 0.1 ^a^**	3.1 ± 0.0 ^f^	65.4 ± 0.1 ^e^	73.3 ± 0.1 ^a^	70.6 ± 0.2 ^b^	60.6 ± 0.0 ^d^	72.2 ± 0.1 ^c^	**74.9 ± 0.1 ^a^**

***Note:*** Values are means ± SD of triplicates (*n* = 3). Gly—glycerol; Xyl—xylose; Ara—arabinose; Fru–fructose; Glu—glucose; Gala—galactose; Suc—sucrose; Mal—maltose; Lac—lactose; Carb—carbohydrate; Tot—total sugars content; BLQ—observed concentration is below limit of quantification; 3 and 6 h—acid hydrolysis of bacterial levan lasted for 3 and 6 h, respectively; DW—the concentration expressed on a dry weight basis; n.d.—not detected. The concentration of carbohydrates in the bacterial levan sample was expressed as g 100 g^−1^ on a dry weight basis. Means within the same carbohydrate and source of exopolysaccharide with different superscript letters (^a^, ^b^, ^c^, ^d^, ^e^, ^f^, and ^g^) are significantly different at *p* < 0.05.

**Table 5 nanomaterials-13-02969-t005:** Antimicrobial activity of colloid systems containing silver nanoparticles according to their diameter of the inhibition zone values.

Test Culture	Average Zone of Inhibition, mm									
	BLss@AgNPs 20		BLlm@AgNPs 20		BLss@AgNPs 50		BLss@AgNPs 500		BLdex@AgNPs 500	
	D	W	D	W	D	W	D	W	D	W
*Pseudomonas aeruginosa* ATCC 27853	10.1 ± 0.0 ^a^	14.1 ± 0.2 ^c^	10.1 ± 0.1 ^a^	14.3 ± 0.1 ^c^	9.2 ± 0.0 ^ab^	17.4 ± 0.2 ^b^	8.1 ± 0.0 ^b^	21.2 ± 0.2 ^a^	9.3 ± 0.1 ^ab^	10.1 ± 0.1 ^d^
*Staphylococcus aureus* ATCC 6538P	10.2 ± 0.0 ^ab^	11.2 ± 0.1 ^b^	10.3 ± 0.1 ^ab^	11.4 ± 0.1 ^b^	11.1 ± 0.1 ^a^	14.2 ± 0.1 ^a^	10.1 ± 0.1 ^ab^	15.5 ± 0.3 ^a^	9.1 ± 0.0 ^b^	12.2 ± 0.2 ^b^
*Escherichia coli* ATCC 25922	9.3 ± 0.0 ^bc^	13.1 ± 0.1 ^c^	10.2 ± 0.0 ^ab^	13.4 ± 0.1 ^c^	10.3 ± 0.1 ^ab^	15.4 ± 0.1 ^b^	11.1 ± 0.1 ^a^	19.3 ± 0.1 ^a^	8.1 ± 0.0 ^c^	10.3 ± 0.2 ^d^
*Enterococcus aerogenes* ATCC 13048	8.1 ± 0.0 ^ab^	12.4 ± 0.0 ^b^	9.1 ± 0.0 ^a^	12.4 ± 0.0 ^b^	8.2 ± 0.0 ^ab^	12.4 ± 0.0 ^b^	9.2 ± 0.0 ^a^	15.1 ± 0.2 ^a^	7.2 ± 0.0 ^b^	6.1 ± 0.0 ^c^
*Pseudomonas aeruginosa* *	9.4 ± 0.0 ^a^	6.1 ± 0.0 ^a^	9.1 ± 0.0 ^a^	6.2 ± 0.0 ^a^	9.3 ± 0.0 ^a^	6.1 ± 0.0 ^a^	9.4 ± 0.0 ^a^	6.2 ± 0.0 ^a^	9.4 ± 0.1 ^a^	6.2 ± 0.0 ^a^
*Staphylococcus epidermidis* *	9.1 ± 0.1 ^b^	11 ± 0.0 ^c^	10.2 ± 0.2 ^ab^	13.3 ± 0.1 ^b^	11.1 ± 0.1 ^a^	15.1 ± 0.1 ^a^	10.2 ± 0.0 ^ab^	16.2 ± 0.1 ^a^	9.3 ± 0.1 ^b^	11.1 ± 0.1 ^c^
*Escherichia coli* *	10.2 ± 0.1 ^a^	6.2 ± 0.0 ^d^	10.1 ± 0.2 ^a^	9.2 ± 0.2 ^c^	11.1 ± 0.0 ^a^	11.3 ± 0.1 ^b^	11.2 ± 0.1 ^a^	14.4 ± 0.2 ^a^	10.1 ± 0.1 ^a^	6.1 ± 0.0 ^d^
*Enterococcus faecium* *	9.3 ± 0 ^b^	8.2 ± 0.0 ^c^	9.2 ± 0.0 ^b^	11.2 ± 0.0 ^b^	11.3 ± 0.1 ^a^	13.1 ± 0.0 ^a^	8.1 ± 0.0 ^b^	14.3 ± 0.1 ^a^	9.4 ± 0.0 ^b^	8.1 ± 0.0 ^c^

***Note:*** Values are expressed as means ± SD values of duplicates (*n* = 2). Means within the same test culture and testing method with different superscript letters (^a^, ^b^, ^c^, and ^d^) are significantly different at *p* ≤ 0.05. * Isolate from clinical samples. The capital letter **D** refers to antimicrobial susceptibility testing by the disc diffusion method. The capital letter **W** refers to antimicrobial susceptibility testing agar by the well diffusion method. Disc diameter of 5 mm; agar well diameter of 6 mm; BLss@AgNPs 20—colloid system containing silver nanoparticles biosynthesized using 20 mg of bacterial levan from *Streptococcus salivarius* K12; BLss@AgNPs 50—colloid system containing silver nanoparticles biosynthesized using 50 mg of bacterial levan from *Streptococcus salivarius* K12; BLss@AgNPs 500—colloid system containing silver nanoparticles biosynthesized using 500 mg of bacterial levan from *Streptococcus salivarius* K12; BLlm@AgNPs 20—colloid system containing silver nanoparticles biosynthesized using 20 mg of bacterial levan from *Leuconostoc mesenteroides* DSM 20343; BLdex@AgNPs 500—colloid system containing silver nanoparticles biosynthesized using 500 mg of high-molecular-weight dextran (Sigma-Aldrich).

**Table 6 nanomaterials-13-02969-t006:** Minimal inhibitory (MIC) and bactericidal (MBC) concentration values of colloid systems containing silver nanoparticles (µL from stock) against selected pathogenic bacteria.

Test Culture	Average MBC and MIC, µL mL^−1^ AgNPs									
	BLss@AgNPs 20		BLlm@AgNPs 20		BLss@AgNPs 50		BLss@AgNPs 500		BLdex@AgNPs 500	
	MBC (95%)	MIC	MBC (95%)	MIC	MBC (95%)	MIC	MBC (95%)	MIC	MBC (95%)	MIC
*Pseudomonas aeruginosa* ATCC 27853	5.2 ± 0.1 ^a^	2.6 ± 0.0 ^b^	5.2 ± 0.1 ^a^	5.2 ± 0.0 ^a^	5.2 ± 0.0 ^a^	5.2 ± 0.0 ^a^	5.2 ± 0.0 ^a^	5.2 ± 0.4 ^a^	2.6 ± 0.1 ^b^	2.6 ± 0.0 ^b^
*Staphylococcus aureus* ATCC 6538P	41.7 ± 0.3 ^b^	20.8 ± 0.3 ^b^	41.7 ± 0.7 ^b^	20.8 ± 0.1 ^b^	20.8 ± 0.3 ^c^	20.8 ± 0.4 ^b^	83.3 ± 2.4 ^a^	83.3 ± 2.1 ^a^	41.7 ± 0.6 ^b^	20.8 ± 0.8 ^b^
*Escherichia coli* ATCC 25922	5.2 ± 0.0 ^b^	5.2 ± 0.0 ^a^	10.4 ± 0.1 ^a^	2.6 ± 0.0 ^b^	2.6 ± 0.0 ^c^	2.6 ± 0.0 ^b^	10.4 ± 0.4 ^a^	5.2 ± 0.3 ^a^	2.6 ± 0.0 ^c^	2.6 ± 0.0 ^b^
*Enterococcus aerogenes* ATCC 13048	20.8 ± 0.4 ^a^	10.4 ± 0.1 ^b^	10.4 ± 0.0 ^b^	10.4 ± 0.1 ^b^	10.4 ± 0.0 ^b^	10.4 ± 0.1 ^b^	10.4 ± 0.3 ^b^	10.4 ± 0.4 ^b^	20.8 ± 0.8 ^a^	20.8 ± 0.7 ^a^
*Pseudomonas aeruginosa* *	5.2 ± 0.0 ^a^	5.2 ± 0.1 ^a^	5.2 ± 0.0 ^a^	5.2 ± 0.0 ^a^	5.2 ± 0.0 ^a^	5.2 ± 0.0 ^a^	5.2 ± 0.1 ^a^	5.2 ± 0.1 ^a^	5.2 ± 0.1 ^a^	5.2 ± 0.1 ^a^
*Staphylococcus epidermidis* *	20.8 ± 0.2 ^a^	10.4 ± 0.2 ^b^	20.8 ± 0.2 ^a^	10.4 ± 0.1 ^b^	20.8 ± 0.4 ^a^	10.4 ± 0.2 ^b^	20.8 ± 0.9 ^a^	10.4 ± 0.5 ^b^	20.8 ± 0.9 ^a^	10.4 ± 0.3 ^b^
*Escherichia coli* *	5.2 ± 0.0 ^b^	5.2 ± 0.0 ^b^	5.2 ± 0.0 ^b^	5.2 ± 0.0 ^b^	5.2 ± 0.1 ^b^	5.2 ± 0.0 ^b^	10.4 ± 0.4 ^a^	10.4 ± 0.4 ^a^	5.2 ± 0.1 ^b^	5.2 ± 0.1 ^b^
*Enterococcus faecium* *	20.8 ± 0.1 ^a^	5.2 ± 0.0 ^b^	10.4 ± 0.1 ^b^	10.4 ± 0.1 ^a^	10.4 ± 0.2 ^b^	5.2 ± 0.0 ^b^	10.4 ± 0.6 ^b^	10.4 ± 0.4 ^a^	20.8 ± 1.4 ^a^	5.2 ± 0.1 ^b^

***Note:*** Values are expressed as means ± SD values of duplicates (*n* = 2). Means within the same test culture and testing method with different superscript letters (^a^, ^b^, ^c^) are significantly different at *p* ≤ 0.05. * Isolate from clinical samples. Disc diameter of 5 mm; agar well diameter of 6 mm. BLss@AgNPs 20—colloid system containing silver nanoparticles biosynthesized using 20 mg of bacterial levan from *Streptococcus salivarius* K12; BLss@AgNPs 50—colloid system containing silver nanoparticles biosynthesized using 50 mg of bacterial levan from *Streptococcus salivarius* K12; BLss@AgNPs 500—colloid system containing silver nanoparticles biosynthesized using 500 mg of bacterial levan from *Streptococcus salivarius* K12; BLlm@AgNPs 20—colloid system containing silver nanoparticles biosynthesized using 20 mg of bacterial levan from *Leuconostoc mesenteroides* DSM 20343; BLdex@AgNPs 500—colloid system containing silver nanoparticles biosynthesized using 500 mg of high-molecular-weight dextran (Sigma-Aldrich).

## Data Availability

The datasets and analysis of this study are available from the corresponding author upon reasonable request.

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
