# Peer review of "Elaboration of Nanostructured Levan-Based Colloid System as a Biological Alternative with Antimicrobial Activity for Applications in the Management of Pathogenic Microorganisms"

_nanomaterials, 2023, doi:10.3390/nano13222969_

Round 1

Reviewer 1 Report

Comments and Suggestions for Authors

The manuscript "Elaboration of nanostructured levan-based colloid system as a biological alternative with antimicrobial activity for applications in the management of pathogenic microorganisms" reports the synthesis, properties and antimicrobial activity of levan-based colloids biosynthesized using two different bacterial strains, having distinct properties in function of the bacterial strain used. The manuscript is worth to be published in Nanomaterials, but several concerns need to be addressed prior to its publication.

Concerns:

1. Please ensure that every abbreviation is defined in its first appearance, both in the abstract, secondly in the body of the manuscript.

2. Please revise the numeration of all sections and subsections, as all of the numbers look partially or totally incorrect.

3. It would be more appropriate to use the word "biosynthesis" (or related derivations) instead of synthesis. The readers with more chemical background would appreciate the difference.

4. Line 181: add a space between the unit and "x g" in the description of the centrifuge use. Same applies in line 282 after NaNO3

5. Line 191: Should it be frequency instead of fervency?

6. The correct city of Pol-Eko aparatura is "Wodzisław Śląski". Please correct (line 268).

7. Figure 5 has very small subfigures, in which captions are difficult to be read, and the details, to be perceived. It occupies only a fraction of the page. It would be better to change the pattern (for instancte 2x5 distribution, in vertical and using all the page). Same fact applies for figure 9. A 3x6 arrangement instead the current 7x3 would be much more readable, and would provide more information to the readers.

8. Table 4. Firstly, something needs to be done for it to me more readable, to avoid that the lines for Fru, Gly and Gala does not appear in they way they are now. For example, reducing an unit the letter size. Or occupying all the page with a vertical orientation. Same applies for Tables 5 and 6.

9. Figure 8. Improve the legend of the figure and the captions in the graph to have all the information in a more clear manner.

Comments on the Quality of English Language

English is generally fine. Just, very rarely, articles a/an/the are not fully properly used

Author Response

Response to Reviewer's 1 comments

A: The authors would like to thank the Reviewer for carefully checking our manuscript and for valuable comments. In preparing the manuscript, the authors have incorporated most of the changes suggested. The authors refer to them in detail below.

R: 1. Please ensure that every abbreviation is defined in its first appearance, both in the abstract, secondly in the body of the manuscript.

A: The authors appreciate the Reviewer's remark. The authors double-checked each acronym the first time it appeared in the manuscript and provided explanations for all of them. Meanwhile, the explanation for all acronyms is ensured at the end of the manuscript.   

R: 2. Please revise the numeration of all sections and subsections, as all of the numbers look partially or totally incorrect.

A: The authors are sorry for such shortcomings. The authors have revised the numeration within the entire manuscript.    

R: 3. It would be more appropriate to use the word "biosynthesis" (or related derivations) instead of synthesis. The readers with more chemical background would appreciate the difference.

A: The authors have substituted the term "synthesis" with "biosynthesis".  

R: 4. Line 181: add a space between the unit and "x g" in the description of the centrifuge use. Same applies in line 282 after NaNO3 

A: The authors appreciate the Reviewer's remark. The authors have added a space between the unit and "x g" within the entire manuscript.

R: 5. Line 191: Should it be frequency instead of fervency?

A: The authors are sorry for such typos. Yes, this is what the authors intended to write. Thank you!

R: 6. The correct city of Pol-Eko aparatura is "WodzisÅ‚aw ÅšlÄ…ski". Please correct (line 268).

A: The authors included the correct city of Pol-Eko aparatura. Thank you!

R: 7. Figure 5 has very small subfigures, in which captions are difficult to be read, and the details, to be perceived. It occupies only a fraction of the page. It would be better to change the pattern (for instancte 2x5 distribution, in vertical and using all the page). Same fact applies for figure 9. A 3x6 arrangement instead the current 7x3 would be much more readable, and would provide more information to the readers.

A: The figures mentioned have been revised according to the Reviewer's suggestions. Many thanks! 

R: 8. Table 4. Firstly, something needs to be done for it to me more readable, to avoid that the lines for Fru, Gly and Gala does not appear in they way they are now. For example, reducing an unit the letter size. Or occupying all the page with a vertical orientation. Same applies for Tables 5 and 6.

A: The authors are sorry for such shortcomings. The shape of the tables mentioned by the Reviewer has been improved by changing the page from "portrait" to "landscape" orientation. The authors hope this does not contradict the requirements of the journal "Nanomaterials." 

R: 9. Figure 8. Improve the legend of the figure and the captions in the graph to have all the information in a more clear manner.

A: The authors improved the quality of Figure 8, which is currently Figure 7. Thank you!

R: English is generally fine. Just, very rarely, articles a/an/the are not fully properly used.

A: The authors have double-checked the spelling of the manuscript. Much appreciated!

Reviewer 2 Report

Comments and Suggestions for Authors

Dear Authors,

the manuscript is devoted to the current topic of the development of nanoparticles for medical use. However, there are several comments on the manuscript:

Figure 1. You cited a well-known reaction from a school or university program in organic chemistry - the reaction of a silver mirror. I suggest removing it because it is a scientific article, not a textbook.

Figure 2. The drawing is too difficult to understand; I suggest removing it, redrawing it or describing it in the text of the manuscript.

Figure 3. A picture that is too difficult to understand, a photo of a laboratory table with signed labels, is unsuitable for publication in a scientific article in a reputable journal as a laboratory diary. I suggest taking a photo on a neutral background of test tubes with solutions followed by graphic processing.

Figure 8. Similar requirements to Figure 3.

Author Response

Response to Reviewer's 2 comments

A: The authors would like to thank the Reviewer for carefully checking our manuscript and for valuable comments. In preparing the manuscript, the authors have incorporated most of the changes suggested. The authors refer to them in detail below.

R: 1. Figure 1. You cited a well-known reaction from a school or university program in organic chemistry - the reaction of a silver mirror. I suggest removing it because it is a scientific article, not a textbook.

A:  The authors understand the Reviewer's concern. The authors agree with the Reviewer's point regarding the well-known reaction of a silver mirror where the formation of metallic silver is implemented by oxidation-reduction reaction during the interaction of an ammonia solution of silver oxide in the presence of aldehydes. However, the authors intend to show that fructose, which lacks an aldose group, may react with AgNOs as reducing agents following the Lobry de Bruyn–Alberda van Ekenstein transformation. The authors removed from the manuscript the mechanism of metal nanoparticle formation while retaining the description of the process. We believe that the readership of the journal nanoparticles, the ones that are not experts in the field of chemistry, would find this information feasible. 

R: 2. Figure 2. The drawing is too difficult to understand; I suggest removing it, redrawing it or describing it in the text of the manuscript.

A: The authors appreciate the Reviewer's point. The authors have revised Figure 2 and hope this is sufficient to see the information in a more understandable way. We inform the reviewer that each step depicted in Figure 2 can be found incorporated in the Materials and Methods section. There is no need to repeat the same things twice.   

R: 3. Figure 3. A picture that is too difficult to understand, a photo of a laboratory table with signed labels, is unsuitable for publication in a scientific article in a reputable journal as a laboratory diary. I suggest taking a photo on a neutral background of test tubes with solutions followed by graphic processing.

A: The authors have revised Figure 3 according to the Revewer’s suggestion. Thank you. 

R: 4. Figure 8. Similar requirements to Figure 3.

A: The authors appreciate the Reviewer's remark. The authors have revised Figure 8 according to the Reviewer’s suggestion. Thank you.

Round 2

Reviewer 1 Report

Comments and Suggestions for Authors

The concerns given in the previous round have been adequately addressed, and now the manuscript is suitable for publication.

Author Response

The authors would like to thank the reviewer for carefully checking our manuscript and for his valuable comments that allowed to improve the manuscript. 

Reviewer 2 Report

Comments and Suggestions for Authors

Dear Authors,

Unfortunately, none of my comments (except one related to figure 9) have been taken into account. In the manuscript, it is difficult to perceive graphical materials.

Author Response

R: Unfortunately, none of my comments (except one related to figure 9) have been taken into account. In the manuscript, it is difficult to perceive graphical materials

A: With this letter, we would like to express concern regarding the objectivity of the Reviewer's No. 2. assessment. The authors do not understand the reviewer's bias since all concerns raised by the reviewer in the original version of the manuscript were addressed by the authors and indicated by the track changes function. The authors believe that the reviewer ignored them. The authors question the objectivity of this review and, therefore, would like to ask the Editor to give his assessment of the graphic materials presented in the manuscript.

R: 1. Figure 1. You cited a well-known reaction from a school or university program in organic chemistry - the reaction of a silver mirror. I suggest removing it because it is a scientific article, not a textbook.

A: The authors understand the Reviewer's concern. The authors agree with the Reviewer's point regarding the well-known reaction of a silver mirror where the formation of metallic silver is implemented by oxidation-reduction reaction during the interaction of an ammonia solution of silver oxide in the presence of aldehydes. However, the authors intend to show that fructose, which lacks an aldose group, may react with AgNOs as reducing agents following the Lobry de Bruyn–Alberda van Ekenstein transformation. The authors removed from the manuscript the mechanism of metal nanoparticle formation while retaining the description of the process. We believe that the readership of the journal nanoparticles, the ones that are not experts in the field of chemistry, would find this information feasible. 

A2: THE AUTHORS REMOVED THIS FIGURE FROM THE MANUSCRIPT WHILE RETAINING THE DESCRIPTION OF THE REACTION!!! WHAT ELSE NEED TO BE IMPROVED?

R: 2. Figure 2. The drawing is too difficult to understand; I suggest removing it, redrawing it or describing it in the text of the manuscript.

A: The authors appreciate the Reviewer's point. The authors have revised Figure 2 and hope this is sufficient to see the information in a more understandable way. We inform the reviewer that each step depicted in Figure 2 can be found incorporated in the Materials and Methods section. There is no need to repeat the same things twice. 

A1: THE AUTHORS IMPROVED THE SIZE OF FONT IN THE FIGURE!!! WHAT ELSE NEED TO BE IMPROVED?

R: 3. Figure 3. A picture that is too difficult to understand, a photo of a laboratory table with signed labels, is unsuitable for publication in a scientific article in a reputable journal as a laboratory diary. I suggest taking a photo on a neutral background of test tubes with solutions followed by graphic processing. 

A: The authors have revised Figure 3 according to the Revewer’s suggestion. Thank you. 

A1: THE AUTHORS REMOVED THE BACKGROUND (TOOK ON A NEUTRAL BACKGROUND) OF THE FIGURE AND IMPROVED THE OVERALL QUALITY OF IT!!! WHAT ELSE NEED TO BE IMPROVED?

R: 4. Figure 8. Similar requirements to Figure 3.

A: The authors appreciate the Reviewer's remark. The authors have revised Figure 8 according to the Reviewer’s suggestion. Thank you. 

A1: THE AUTHORS REVISED FIGURE 8 BY INCREASING THE LEGEND FONT SIZE, PLOT VALUES, X AND Y SCALING!!! WHAT ELSE NEED TO BE IMPROVED?

Round 3

Reviewer 2 Report

Comments and Suggestions for Authors

Dear Authors,

my suggestions are taken into account.